# Towards Diverse and Faithful One-shot Adaption of Generative Adversarial Networks

**Yabo Zhang** [*,1]  **Mingshuai Yao** [1]  **Yuxiang Wei** [1]  **Zhilong Ji** [2]  **Jinfeng Bai** [2]  **Wangmeng Zuo** [1,3] (✉)

[1]Harbin Institute of Technology    [2]Tomorrow Advancing Life    [3]Peng Cheng Laboratory

## Abstract

One-shot generative domain adaption aims to transfer a pre-trained generator on one domain to a new domain using one reference image only. However, it remains very challenging for the adapted generator (i) to generate diverse images inherited from the pre-trained generator while (ii) faithfully acquiring the domain-specific attributes and styles of the reference image. In this paper, we present a novel one-shot generative domain adaption method, *i.e.*, DiFa, for diverse generation and faithful adaptation. For global-level adaptation, we leverage the difference between the CLIP embedding of reference image and the mean embedding of source images to constrain the target generator. For local-level adaptation, we introduce an attentive style loss which aligns each intermediate token of adapted image with its corresponding token of the reference image. To facilitate diverse generation, selective cross-domain consistency is introduced to select and retain the domain-sharing attributes in the editing latent $\mathcal{W}+$ space to inherit the diversity of pre-trained generator. Extensive experiments show that our method outperforms the state-of-the-arts both quantitatively and qualitatively, especially for the cases of large domain gaps. Moreover, our DiFa can easily be extended to zero-shot generative domain adaption with appealing results. Code is available at `https://github.com/YBYBZhang/DiFa`.

## 1  Introduction

Generative adversarial networks (GANs) [8] have achieved remarkable progress in generating photo-realistic and highly-diverse images [11, 12, 18]. However, GANs usually require a large number of samples for stable training, and suffer from severe mode collapse when trained with insufficient data (*e.g.*, one image). Recently, several works [2, 10, 27, 29, 34, 37, 39] have been proposed to train a GAN from scratch with only one or few images, but are limited in generating high quality and diverse images. In this paper, we resort to one-shot or few-shot generative domain adaption (GDA), *i.e.*, transferring a pre-trained generator on one domain to a new domain using one or few reference images (as shown in Fig. 1). Thus, GDA can provide a new perspective to address the above issues by inheriting the generation ability and diversity of the pre-trained generator.

Many methods [16, 19, 21, 25, 31, 33, 35, 38] have been proposed for one-shot GDA. Nonetheless, domain-specific attributes and styles usually can be described by language, and thus can be well depicted by Contrastive-Language-Image-Pretraining (CLIP) [22]. Hence, CLIP-based one-shot GDA methods [7, 15, 32, 41] have been proposed to adapt a pre-trained generator, *e.g.*, StyleGAN2 [12], to the target domain. In particular, the domain-gap direction between source and target domains is first calculated in the CLIP embedding space. Then, the pre-trained generator is transferred by aligning the CLIP direction between the source and adapted images with the domain-gap direction.

---

[*]The first author performed this work as an intern at Tomorrow Advancing Life.

36th Conference on Neural Information Processing Systems (NeurIPS 2022).

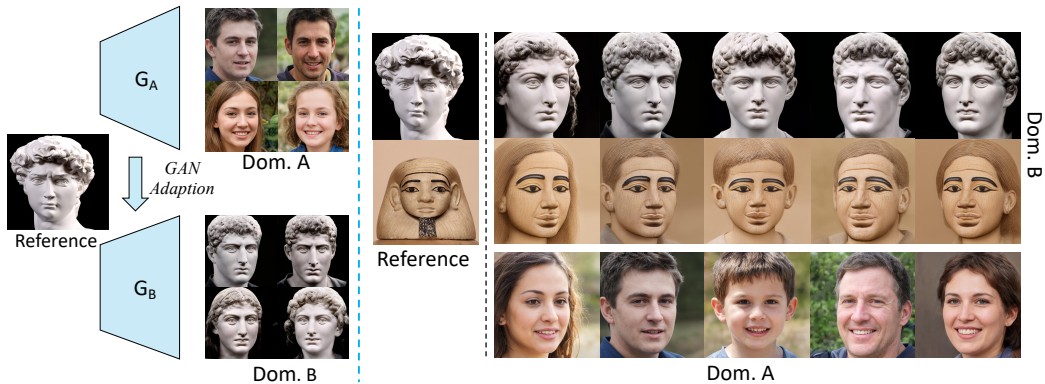

Figure 1: **Diverse and faithful one-shot generative domain adaption. Left:** One-shot generative domain adaption aims to transfer a pre-trained GAN from domain $A$ (*e.g.*, FFHQ) to domain $B$ (*e.g.*, sculptures) by providing one reference image only. **Right:** Synthesised images by our DiFa. Notably, our DiFa is effective in inheriting the diverse generation ability of GAN from domain $A$, while faithfully acquiring the representative characteristics of the reference image in domain $B$.

However, it remains very challenging for the adapted generator (i) to generate images as *diverse* as the pre-trained generator while (ii) *faithfully* acquiring the domain-specific attributes and styles of the reference image. Firstly, although the domain-gap direction extracts the pronounced characteristics of the reference image, the detailed local styles and attributes are usually ignored in CLIP embedding. Without considering these local styles and attributes, the adapted generator cannot faithfully acquire the domain-specific characteristics of the reference image. Secondly, the domain-gap direction is the difference between the CLIP embedding of the reference image and source domain, which contains both domain-specific attribute shifts (*e.g.*, thick eyebrows in second row of Fig. 1) and domain-sharing attribute shifts (*e.g.*, gender). Directly aligning the training sample-shift direction with the domain-gap direction introduces the unnecessary domain-sharing attribute changes to the adapted images, thereby being harmful to inheriting the diversity from the pre-trained generator. Although [15, 41] proposed to use the style mixing and editing direction preservation to address these issues, only limited improvements are achieved.

In this work, we present a novel one-shot GDA method, *i.e.*, DiFa, for diverse generation and faithful adaption. In terms of faithful adaption, we consider both attributes and styles. For global-level adaptation, we define the domain-gap direction as the difference between the CLIP embedding of reference image and the mean embedding of source images. As for local-level adaptation, we introduce an attentive style (AS) loss on the intermediate layer of the CLIP image encoder. For each intermediate token of an adapted image, it first finds the nearest token of the reference image, and then minimizes their difference to make GDA adapt to the target style. In terms of diverse generation, selective cross-domain consistency (SCC) is introduced to select and retain domain-sharing attributes in the editing latent $\mathcal{W}+$ space to inherit the diversity of pre-trained generator. In particular, we use a styleGAN inversion models [24, 30] to invert the images from source and target domains into the $\mathcal{W}+$ space. Then, we compute the direction $\Delta \boldsymbol{w}$ between the two domains, where smaller values in $\Delta \boldsymbol{w}$ indicate that the corresponding latent variables in $\mathcal{W}+$ space are domain-sharing attributes. Selective cross-domain consistency encourages an adapted image and its corresponding source image to be similar in domain-sharing attributes, and can be different in other attributes. SCC allows the adapted generator to inherit from the pre-trained generator selectively. Thus our DiFa can guarantee the diversity of adapted images without the sacrifice of decreasing domain adaption ability.

Quantitative and qualitative experiments are conducted on a wide range of source and target domains. Evaluation results highlight the superiority of our DiFa compared against the state-of-the-art methods, especially for the cases of large domain gaps (*e.g.*, Cat → Tiger). To illustrate the editing capabilities of the adapted latent space, we employ InterFaceGAN [28] to edit the real images in target domain.

Overall, our contributions are summarized as follows:

- We introduce a novel method namely DiFa, along with selective cross-domain consistency and attentive style losses, for diverse generation and faithful adaption.

- Extensive experiments show the effectiveness of our DiFa in acquiring the representative domain characteristics from the reference image, and inheriting the ability of pre-trained generator to produce diverse images.
- Our DiFa can be easily extended to zero-shot generative domain adaption with appealing results.

## 2 Related Work

**Few-shot Domain Adaption of GANs.** Few-shot generative domain adaption aims to transfer a generator pre-trained on a source domain to a new target domain with very limited reference images. Earlier studies [16, 19, 21, 25, 31, 33, 35, 38] utilized the adversarial loss [8] to capture domain-specific information from given reference images. To reduce mode collapse, these methods usually adopted fewer learnable parameters [19, 25, 31, 35] or introduce regularization terms [16, 21, 33, 38], but still produce images with insufficient diversity. With the success of CLIP [22], recent works [7, 15, 32, 41] leveraged the difference between the CLIP embeddings of the source and target domains to guide the attribute-level adaption, beating the methods with adversarial loss [8]. To better capture domain-specific styles, several methods [15, 41] adopted the style mixing trick during inference time, however, it may bring undesired semantic artifacts when there is a significant shape discrepancy. [15, 32, 41] attempted to generate diverse images by preserving the editing distance of input pairs, before and after adaption. Nonetheless, they indistinguishably retain both domain-sharing and domain-specific attributes, which is conflicted with faithful adaption.

**GAN Inversion.** GAN inversion aims to invert an image into its corresponding latent codes, which can be grouped into optimization-based and encoder-based methods. Optimization-based inversion [5, 40] directly updates the latent code by minimizing the reconstruction error. Albeit high-quality and accurate reconstruction can be obtained, it usually costs a few minutes for an image. In contrast, encoder-based algorithms [1, 24, 30] directly embed a given image into latent codes, so that the inference can be completed in real-time, and the gradients of input images could also be passed backward. Moreover, encoder-based algorithms also achieve considerable performance when handling out-of-domain images, and thus it is feasible to project adapted images into $\mathcal{W}^+$ codes during training.

## 3 Proposed Method

In this work, we focus on one-shot generative domain adaption task, which aims to transfer a generator $G_A$ pre-trained on domain $A$ to a new domain $B$ using one reference image $I_{tar}$ only. Specifically, we present a novel method termed DiFa to generate diverse images inherited from the pre-trained generator while faithfully acquiring the domain-specific attributes and styles of the reference image. The overview of our DiFa is illustrated in Fig. 2. In this section, we first introduce the global-level adaption loss with an estimated domain-gap direction. The attentive style loss and selective cross-domain consistency loss are then proposed for local-level adaptation and diverse generation, respectively. Finally, we introduce the overall learning objective for training.

### 3.1 Global-level Adaption

Recent studies [7, 15, 41] have demonstrated the superiority of CLIP in transferring a generator in one domain to a new domain under the one-shot setting. In comparison to the methods based on adversarial loss [16, 19, 21, 25, 31, 33, 35, 38], CLIP-based methods are effective in describing domain characteristics and resulting in photo-realistic images. Given a generator $G_A$ pre-trained on domain $A$ and a target reference image $I_{tar}$ from domain $B$, CLIP-based methods first calculate the domain-gap direction between domain $B$ and $A$:

$$\Delta \boldsymbol{v}_{dom} = \boldsymbol{v}_{tar} - \boldsymbol{v}_{src}, \tag{1}$$

where $\boldsymbol{v}_{tar} = E_I(I_{tar})$ denotes the embedding of target domain $B$ and $E_I$ is the CLIP image encoder. $\boldsymbol{v}_{src}$ represents the CLIP embedding of source domain $A$. To transfer $G_A$ to domain $B$, they copy a new generator $G_B$ from $G_A$ and finetune it by aligning the sample-shift direction $\Delta \boldsymbol{v}_{samp}$ with the domain-gap direction $\Delta \boldsymbol{v}_{dom}$:

$$\Delta \boldsymbol{v}_{samp} = \boldsymbol{v}_B - \boldsymbol{v}_A, \tag{2}$$

$$\mathcal{L}_{global} = 1 - \frac{\Delta \boldsymbol{v}_{samp} \cdot \Delta \boldsymbol{v}_{dom}}{|\Delta \boldsymbol{v}_{samp}||\Delta \boldsymbol{v}_{dom}|}, \tag{3}$$

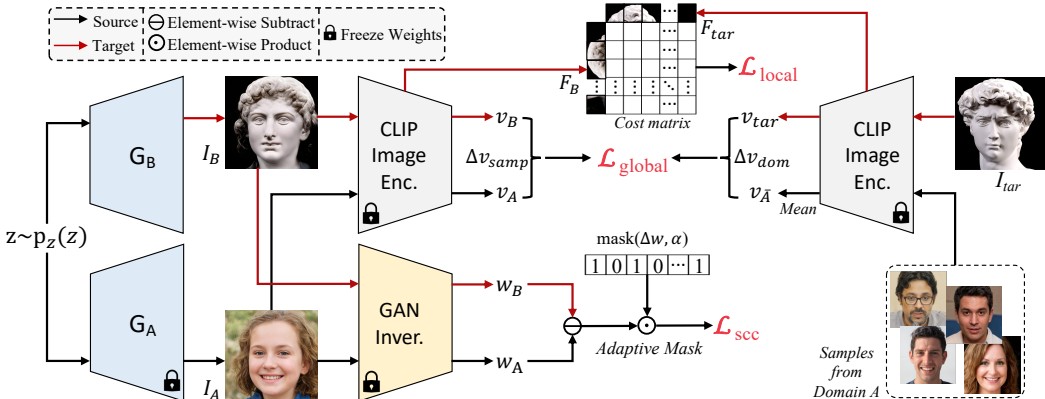

Figure 2: **Overview of our DiFa.** The adapted generator $G_B$ is initialized by pre-trained generator $G_A$. With the aid of CLIP image encoder, the global-level adaption loss $\mathcal{L}_{global}$ and attentive style loss $\mathcal{L}_{local}$ encourage $G_B$ to faithfully acquire both global and local representative domain-specific characteristics from the reference $I_{tar}$. To facilitate diverse generation inherited from $G_A$, the selective cross-domain consistency loss $\mathcal{L}_{scc}$ selects and retains domain-sharing attributes.

where $\boldsymbol{v}_B = E_I(G_B(\boldsymbol{z}))$ and $\boldsymbol{v}_A = E_I(G_A(\boldsymbol{z}))$ denote the CLIP embeddings of domain $B$ and domain $A$ samples. $\boldsymbol{z} \sim \mathcal{N}(0, I)$ denotes the input noise. After finetuning with the global-level adaption loss $\mathcal{L}_{global}$, the adapted generator $G_B$ can generate high-quality images of domain $B$.

Note that the embedding of source domain $\boldsymbol{v}_{src}$ can be calculated in different ways. Two-stage methods [15,41] find the image corresponding to $I_{tar}$ in domain $A$ and treat its CLIP-space embedding as $\boldsymbol{v}_{src}$. Nonetheless, the corresponding image in domain $A$ inevitably contains domain-specific attributes of $I_{tar}$, leading to $\Delta\boldsymbol{v}_{dom}$ ignoring these domain-specific attributes. One-stage methods [7] utilize the mean embedding of source images as $\boldsymbol{v}_{src}$. Intuitively, the mean embedding usually represents the common attributes of source domain, and does not affect domain-specific attribute shifts in $\Delta\boldsymbol{v}_{dom}$. Thus, in our experiments, we use the mean embedding of source images as the source domain embedding, *i.e.*, $\boldsymbol{v}_{src} = \boldsymbol{v}_{\bar{A}} = \mathbb{E}_{\boldsymbol{z}\sim\mathcal{N}(0,I)}[E_I(G_A(\boldsymbol{z}))]$.

### 3.2 Local-level Adaption

Albeit the domain-gap direction $\Delta\boldsymbol{v}_{dom}$ captures the global-level representative domain characteristics of reference image $I_{tar}$, the local attributes and visual styles are usually ignored in CLIP embedding. Therefore, training $G_B$ with $\mathcal{L}_{global}$ only cannot faithfully capture the local-level domain-specific characteristics of $I_{tar}$. For example, images generated by StyleGAN-NADA [7] fail to acquire the mane and stripes of tigers during the Cat $\rightarrow$ Tiger adaption (the first row in Fig. 6(c)). [15,41] tried to inherit the detailed visual styles from $I_{tar}$ through the style mixing. However, when there is a significant shape discrepancy (*e.g.*, pose or cross-category) between $I_{tar}$ and the original adapted image $I_B$, the mismatch of content and style in $I_B^{mix}$ will lead to visible artifacts (the last row in Fig. 5(b) and the first row in Fig. 6(b)).

To mitigate the above issue, we further present an attentive style loss $\mathcal{L}_{local}$ to help $G_B$ faithfully acquire the local-level representative attributes and styles of $I_{tar}$. Inspired by content-style alignment in style transfer [14], $\mathcal{L}_{local}$ is designed to encourage each part of $I_B$ to attentively align with its corresponding styles from $I_{tar}$. Specifically, we first extract the intermediate tokens of $I_B$ and $I_{tar}$ from the $k$-th layer of CLIP image encoder (shown in Fig. 4), and then align each of adapted tokens $\boldsymbol{F}_B$ with its closest target token from $\boldsymbol{F}_{tar}$, where $\boldsymbol{F}_B = \{\boldsymbol{F}_B^1, \ldots, \boldsymbol{F}_B^n\}$ and $\boldsymbol{F}_{tar} = \{\boldsymbol{F}_{tar}^1, \ldots, \boldsymbol{F}_{tar}^m\}$ are the extracted tokens. The final attentive style loss is defined as,

$$\mathcal{L}_{local} = \max\left(\frac{1}{n}\sum_i \min_j \boldsymbol{C}_{i,j}, \frac{1}{m}\sum_j \min_i \boldsymbol{C}_{i,j}\right), \tag{4}$$

where $\boldsymbol{C}$ is the cost matrix to measure the token-wise distances from $F_B$ to $F_{tar}$, and each element of $\boldsymbol{C}$ is computed as:

$$\boldsymbol{C}_{i,j} = 1 - \frac{\boldsymbol{F}_B^i \cdot \boldsymbol{F}_{tar}^j}{|\boldsymbol{F}_B^i||\boldsymbol{F}_{tar}^j|}. \tag{5}$$

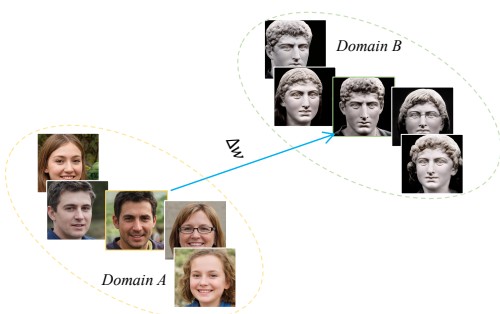

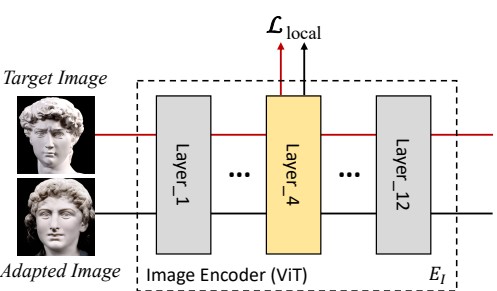

Figure 3: **The process of computing $\Delta w$ between domain $A$ and domain $B$.** Given two clusters of $\mathcal{W}+$ codes from domain $A$ and domain $B$, $\Delta \boldsymbol{w}$ is defined as the difference of their cluster centers.

Figure 4: **Illustration of the Attentive Style loss.** We extract the intermediate adapted and target tokens from the Layer_4 of CLIP image encoder, following by aligning each adapted token with its corresponding target token attentively.

### 3.3 Selectively Diverse Generation

The ability to generate diverse target domain images is also critical for one-shot generative domain adaption. Recall that the domain-gap direction is the difference between the embedding of $I_{tar}$ and source domain, which contains both domain-specific and domain-sharing attribute shifts. Training $G_B$ with $\mathcal{L}_{local}$ also introduces the unnecessary domain-sharing attribute changes to the adapted images, which hinders $G_B$ from inheriting the diversity of the pre-trained generator $G_A$. To facilitate diverse generation, we propose a selective cross-domain consistency loss to select and retain the domain-sharing attributes in $\mathcal{W}+$ space. Intuitively, if an attribute is similar between domains $A$ and $B$ during adaption, it is more likely to be a domain-sharing attribute. According to this assumption, we can dynamically analyze and preserve the domain-sharing attributes. Specifically, we first invert $G_A(\boldsymbol{z})$ and $G_B(\boldsymbol{z})$ into $\mathcal{W}+$ latent codes $\boldsymbol{w}_A$ and $\boldsymbol{w}_B$ with an pre-trained inversion model (*e.g.*, pSp [24] or e4e [30]) for each iteration. Then, as shown in Fig. 3, we compute the difference $\Delta \boldsymbol{w}$ between the centers of a queue of $\mathcal{W}+$ latent codes $\mathcal{X}_A$ and a queue of $\mathcal{W}+$ latent codes $\mathcal{X}_B$, where $\mathcal{X}_A$ and $\mathcal{X}_B$ are dynamically updated with $\boldsymbol{w}_A$ and $\boldsymbol{w}_B$ during training. According to $\Delta \boldsymbol{w}$, we encourage $\boldsymbol{w}_A$ and $\boldsymbol{w}_B$ to be consistent in channels with less difference,

$$\mathcal{L}_{scc} = ||\text{mask}(\Delta \boldsymbol{w}, \alpha) \cdot (\boldsymbol{w}_B - \boldsymbol{w}_A)||_1, \tag{6}$$

where $\alpha$ represents the proportion of preserved attributes and $\text{mask}(\Delta \boldsymbol{w}, \alpha)$ determines which channels to be retained. Let $|\Delta \boldsymbol{w}_{s_{\alpha N}}|$ be the $\alpha N$-th largest element of $|\Delta \boldsymbol{w}|$, and each dimension of $\text{mask}(\Delta \boldsymbol{w}, \alpha)$ is calculated as:

$$\text{mask}(\Delta \boldsymbol{w}, \alpha)_i = \begin{cases} 1 & |\Delta \boldsymbol{w}_i| < |\Delta \boldsymbol{w}_{s_{\alpha N}}| \\ 0 & |\Delta \boldsymbol{w}_i| \geq |\Delta \boldsymbol{w}_{s_{\alpha N}}| \end{cases}. \tag{7}$$

Note that fine layers of StyleGAN [11, 12] usually control color information, and constraining them may have a detrimental effect on obtaining styles of $I_{tar}$. Hence, we only use latent codes corresponding to coarse spatial resolutions ($4^2$–$8^2$) and middle resolutions ($16^2$–$32^2$) in $\mathcal{L}_{scc}$.

### 3.4 Overall Training Loss

Our overall training loss consists of three terms, *i.e.*, the global-level adaption loss $\mathcal{L}_{global}$, the attentive style loss $\mathcal{L}_{local}$ for acquiring detailed style information and the selective cross-domain consistency loss $\mathcal{L}_{scc}$ for inheriting the diversity:

$$\mathcal{L}_{overall} = \mathcal{L}_{global} + \lambda_{local}\mathcal{L}_{local} + \lambda_{scc}\mathcal{L}_{scc}. \tag{8}$$

In our experiments, we use $\lambda_{local} = 2$ and $\lambda_{scc} = \max(0, \frac{n_{iter}-n_B}{N_{iter}-n_{iter}})$, where $N_{iter}$ and $n_{iter}$ denote the total number of training iterations and the $n_{iter}$-th iteration of training, respectively. That is, $\lambda_{scc}$ increases linearly as the training proceeds.

## 4 Experiments

In this section, we first introduce the experimental settings of our DiFa, including implementation details, datasets, and metrics (Sec. 4.1). Both qualitative and quantitative experiments are conducted

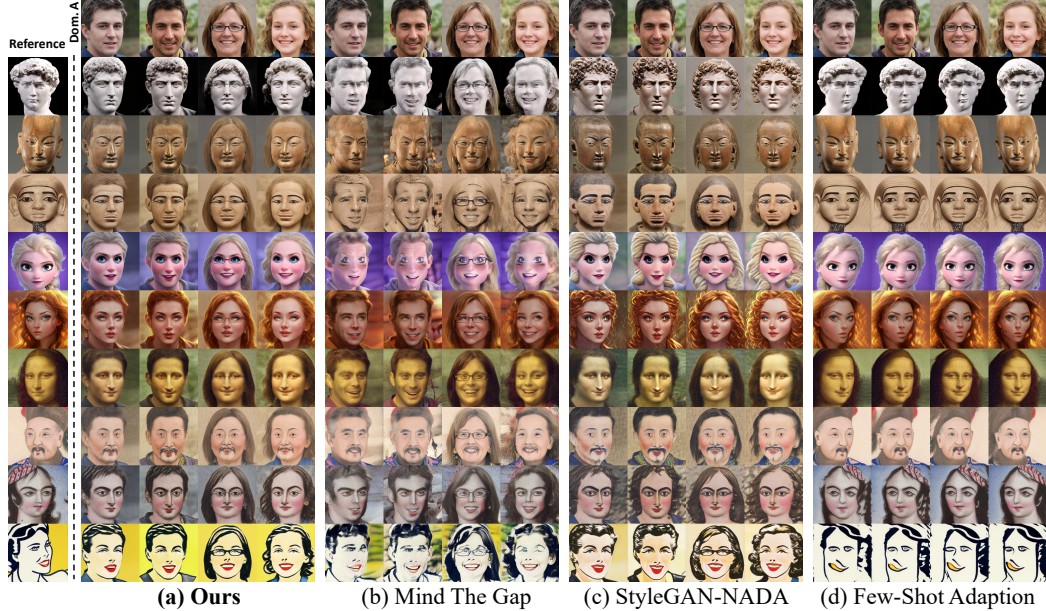

**(a) Ours**   (b) Mind The Gap   (c) StyleGAN-NADA   (d) Few-Shot Adaption

Figure 5: **Qualitative comparisons using the generator pre-trained on FFHQ** [11] between our DiFa, Mind The Gap [41], StyleGAN-NADA [7] and Few-Shot Adaption [21]. The first row and first column show source images in domain $A$ and reference images in domain $B$. Our DiFa not only inherits the ability from the pre-trained generator to produce highly diverse and photo-realistic images, but also faithfully acquires the representative characteristics from the reference images, significantly outperforming the competing methods. **Results best seen at 500% zoom.**

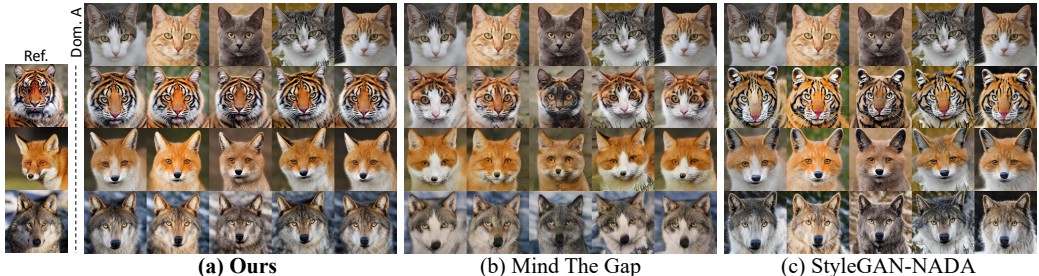

**(a) Ours**   (b) Mind The Gap   (c) StyleGAN-NADA

Figure 6: **Qualitative comparisons using the generator pre-trained on AFHQ-Cat [4].** The first row shows source images in domain $A$ while the first column presents reference images in domain $B$. Our DiFa captures both the representative attributes and styles from various categories of reference images, and exhibits better performance in comparison to the competing methods. In contrast, StyleGAN-NADA [7] misses some domain-specific styles while Mind The Gap [41] fails to obtain the essential attributes of animals in domain $B$. **Results best seen at 500% zoom.**

on a wide range of domains to demonstrate the superiority of our DiFa in generating diverse images and faithful adaption (Sec. 4.2 and Sec. 4.3). Besides, ablation studies are considered to evaluate the effects of our proposed two losses (Sec. 4.4). Finally, we also investigate the editing ability of the adapted generator and extend our DiFa to zero-shot generative domain adaption (Sec. 4.5).

## 4.1   Experimental Settings

**Implementation Details.**   In our experiments, we use StyleGAN2 pre-trained on FFHQ [11] and StyleGAN-ADA [10] pre-trained on AFHQ-Cat [4], and employ e4e [30] and pSp [24] as their inversion models, respectively. Following StyleGAN-NADA [7], we utilize both ViT-B/16 and ViT-B/32 [6] models for CLIP-base losses. For training, we use ADAM optimizer [13] with a learning rate 0.02 and set the batch size to 2. We finetune the generator for 300∼400 iterations, which takes about 3∼4 minutes on an RTX 2080Ti GPU.

Table 1: **KID ($\downarrow$) comparisons between different one-shot domain adaption methods.** Each result is averaged over 5 training shots and in the form of {mean $\pm$ standard error}. FSA, NADA and MTG denote Few-Shot Adaption [21], StyleGAN-NADA [7] and Mind The Gap [41], respectively.

| Models | FFHQ | | | Cat | | |
|---|---|---|---|---|---|---|
| | Amedeo. | Fernand. | Raphael | Tiger | Fox | Wolf |
| FSA [21] | 180.10$\pm$ 1.12 | 187.26$\pm$ 13.10 | 165.25$\pm$ 66.31 | - | - | - |
| NADA [7] | 131.03$\pm$ 28.14 | 169.83$\pm$ 31.52 | 149.19$\pm$ 55.91 | 13.83$\pm$ 2.75 | 73.17$\pm$ 39.30 | 47.96$\pm$ 20.37 |
| MTG [41] | 146.84$\pm$ 46.24 | 192.19$\pm$ 42.73 | 125.58$\pm$ 18.63 | 48.27$\pm$ 13.87 | 69.19$\pm$ 26.23 | 51.11$\pm$ 11.32 |
| **Ours** | **121.21 $\pm$ 24.62** | **159.93 $\pm$ 31.39** | **112.72 $\pm$ 17.61** | **13.13 $\pm$ 2.09** | **54.20 $\pm$ 31.42** | **33.52 $\pm$ 9.21** |

Table 2: **FID ($\downarrow$) comparisons between different one-shot domain adaption methods.** Each result is averaged over 5 training shots and in the form of {mean $\pm$ standard error}.

| Models | FFHQ | | | Cat | | |
|---|---|---|---|---|---|---|
| | Amedeo. | Fernand. | Raphael | Tiger | Fox | Wolf |
| FSA [21] | **171.56 $\pm$ 33.68** | **236.61$\pm$ 25.03** | 177.47$\pm$ 32.21 | - | - | - |
| NADA [7] | 188.44$\pm$ 19.15 | 257.27$\pm$ 19.39 | 186.20$\pm$ 28.60 | 16.74 $\pm$ 1.53 | 82.59 $\pm$ 25.31 | 54.28$\pm$ 13.34 |
| MTG [41] | 215.88$\pm$ 34.14 | 278.46$\pm$ 48.27 | 193.76$\pm$ 7.07 | 46.72$\pm$ 13.34 | 82.30$\pm$ 15.09 | 58.65$\pm$ 6.60 |
| **Ours** | 187.28 $\pm$ 24.45 | 254.68 $\pm$ 17.73 | **172.34 $\pm$ 10.15** | **16.26 $\pm$ 1.08** | **71.57 $\pm$ 18.18** | **44.39 $\pm$ 5.96** |

**Datasets.** For FFHQ adaption, the target images are collected from three datasets: (i) Artstation-Artistic-face-HQ (AAHQ) [17], (ii) MetFaces [10], and (iii) face paintings by Amedeo Modigliani, Fernand Leger and Raphael [36]. Each of them contains 10 images. For Cat adaption, we collect target images from the AFHQ-Wild validation dataset and divide them into Tiger, Fox, and Wolf datasets, which include 103, 53, and 46 images, respectively. In particular, Amedeo Modigliani, Fernand Leger, Raphael, Tiger, Fox and Wolf are used in quantitative experiments.

**Metrics.** Following StyleGAN-ADA [10], we use Fréchet Inception Distance (FID) [9] and Kernel Inception Distance (KID) [3] to evaluate our DiFa quantitatively. Both metrics measure the quality and diversity of the images, while KID is more suitable for the few-shot setting (only a few images in validation sets). In all our experiments, both FID and KID are calculated between 5,000 synthesized images and each validation sets.

## 4.2 Qualitative and Quantitative Evaluation

**Qualitative Results.** Fig. 5 shows the qualitative comparisons adapted from FFHQ [11]. As shown in the figure, Few-Shot Adaption [21] suffers from severe model collapse and generates similar images. Due to StyleGAN-NADA [7] is trained by aligning the sample-shift direction $\Delta v_{samp}$ with domain-gap direction $\Delta v_{dom}$, which contains the domain-sharing attributes (*e.g.*, gender) shift, it also cannot inherit the sufficient diversity from the pre-trained generator. For example, the gender of adapted images is changed to female in 4$\sim$6-th rows in Fig. 5(c). Mind The Gap [41] retains the local styles of reference image via style mixing. However, it produces undesired semantic artifacts when there is a significant shape discrepancy between domains, *e.g.*, redundant noses and eyes in 3rd and last row of Fig. 5(b). In contrast, with the proposed SCC and AS losses, our DiFa not only faithfully acquires the representative domain-specific attributes and styles from the reference image, but also produces images with high diversity inherited from the pre-trained generator. Additionally, we also illustrate the qualitative results adapted from AFHQ-Cat [4] in Fig. 6. Our DiFa also captures sufficient domain-specific characteristics from the reference image in comparison to the competing methods (*e.g.*, the mane and stripes of the tiger in first row), further demonstrating the superiority of our method. More visualizations adapted from other domains are shown in *Suppl*.

**Quantitative Results.** We also quantitatively compare our DiFa with competing methods [7, 21, 41] under six settings, *i.e.*, FFHQ $\rightarrow$ {Amedeo Modigliani, Fernand Leger, Raphael} and Cat $\rightarrow$ {Tiger, Fox, Wolf}. For each setting, we randomly sample an image from a target dataset to perform adaption, and report both Kernel Inception Distance (KID) [3] and Fréchet Inception Distance (FID) [9] metrics. To reduce random sampling error, we repeat it five times and use the mean value as final score. The results are listed in Table 1 and Table 2. One can see that our DiFa clearly outperforms the competing methods, which are consistent with qualitative results in Fig. 5 and Fig. 6. We observe

Table 3: **User preference study.** The numbers represent the percentage of users who favor the images synthesized by our DiFa over the other competitor.

| Model Comparison | Image Quality | Style Similarity | Attribute Consistency |
|---|---|---|---|
| Ours vs. FSA [21] | 87.90% | 36.00% | 96.10% |
| Ours vs. NADA [7] | 76.76% | 77.33% | 78.95% |
| Ours vs. MTG [41] | 81.43% | 73.62% | 64.05% |

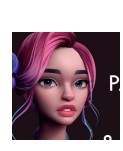 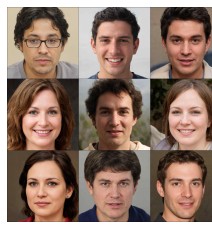 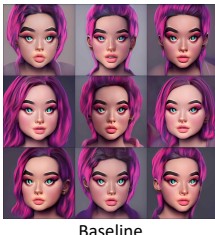 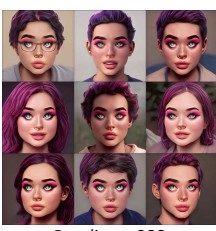 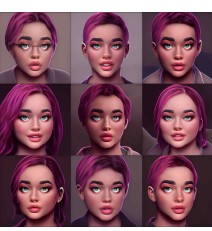

Reference   Dom. A   Baseline   Baseline + SCC  Baseline + SCC + AS (Ours)

Figure 7: **Ablation studies on Selective Cross-modal Consistency (SCC) loss and Attentive Style (AS) loss**. Compared to the baseline, SCC enhances keeping the consistency (*e.g.*, hairstyles) between source and adapted images, vastly boosting the diversity of generation. Moreover, AS encourages the adapted generator to further acquire the representative styles (*e.g.*, darker hue and purple hair) from the reference.

that FSA [21] obtains better FID scores in Amedeo and Fernand datasets, which is inconsistent with above qualitative results (see Fig. 5(d)). Note that FID cannot reflect the overfitting problem very well when target dataset is extremely small and biased [10]. Specifically, these two small datasets have different data biases with FFHQ, *e.g.*, gender bias. 8/10 images in the Amedeo Modigliani dataset and 9/10 images in the Fernand Leger dataset are female. Due to our DiFa acquiring the diversity from the original generator which is trained on FFHQ, it generates male and female adapted images with similar probability. In contrast, for FSA, the adapted images are all similar to the reference image. When comparing on the above two datasets, FSA tends to generate images that have similar gender distribution to the validation dataset, thus achieving better FID results. For the Raphael dataset, which has 5/10 images that are female, our DiFa achieves better FID results.

## 4.3 User Study

We further perform user study to compare our DiFa with the competing methods. Specifically, we provide users a reference image, a source image, and two adapted images from different methods, and ask them to choose the better adapted image for each of three measurements: (i) image quality, (ii) style similarity with the reference and (iii) attribute consistency with the source image. We randomly generate 1,050 samples for each comparison (3,150 in total). There are 30 users. We assign 105 samples for each of them, and give them unlimited time to complete the evaluation. From Table 3, the users strongly favor our DiFa in all three aspects, especially from the perspective of image quality and attribute consistency. Note that FSA [21] suffers from severe mode collapse and simply copies from the reference, hence, it is favored on style similarity but performs worse on the other aspects.

## 4.4 Ablation Study

Ablation studies are conducted to evaluate the effects of two critical components of our DiFa, *i.e.*, the selective cross-domain consistency (SCC) loss and the attentive style (AS) loss. As shown in Fig. 7, the images from the baseline have very limited diversity and lack some representative characteristics of the reference image, *e.g.*, darker hue. Benefited from SCC, the adapted generator begins to retrain the domain-sharing attributes (*e.g.*, hair length and gender), thereby inheriting the diverse generation ability from the pre-trained generator. When further adding AS, we observe that the adapted generator faithfully captures the domain-specific styles and local-level representative attributes from the reference image, *e.g.*, darker hue and purple hair. More ablation studies about hyper-parameters are provided in the *Suppl*.

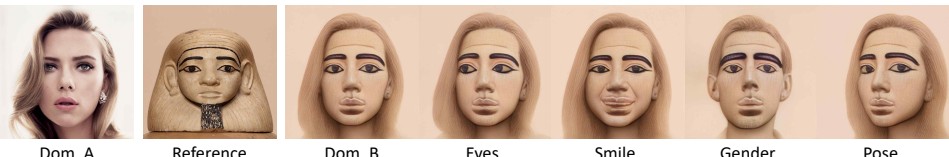

| Dom. A | Reference | Dom. B | Eyes | Smile | Gender | Pose |

Figure 8: **Editing a real image in domain** $B$**.** The first three columns show a real image in domain $A$, a reference image, and an adapted real image in domain $B$, respectively. The other columns present the editing operations and their corresponding results in domain $B$. All editing directions are discovered by InterfaceGAN [28] in domain $A$.

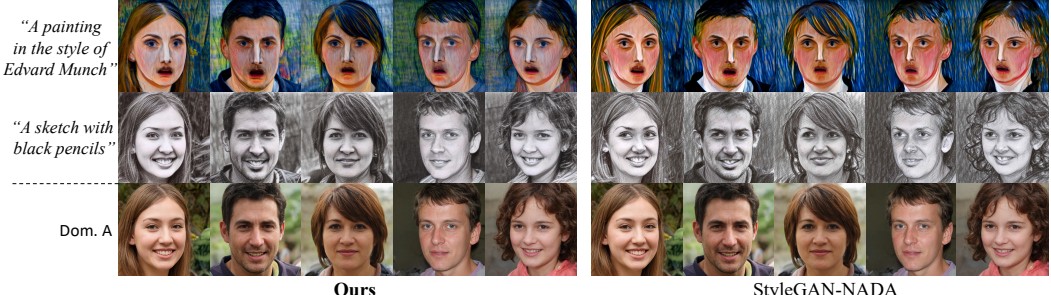

"A painting in the style of Edvard Munch"

"A sketch with black pencils"

Dom. A

**Ours**          StyleGAN-NADA

Figure 9: **Qualitative comparisons on zero-shot generative domain adaption** between our DiFa and StyleGAN-NADA [7]. Given the text description in first column, our DiFa adapts source images in domain $A$ (last row) to the described target domain, significantly surpassing StyleGAN-NADA [7] from the consistency perspective.

## 4.5 Extensions

**Latent Space Editing.** In Fig. 8, we illustrate the editing results performed on a real image adapted into a new domain. Concretely, we employ InterfaceGAN [28] to discover some editing directions in domain $A$, and then leverage these directions to edit the adapted real image. As can be seen, the directions from domain $A$ still manage to control real images in domain $B$, indicating that the adapted generator maintains a similar ability in latent-based editing with the original generator.

**Zero-shot Domain Adaption of GANs.** With minor modifications (*e.g.*, removing the AS loss), our DiFa can be easily extended to zero-shot GDA, *i.e.*, adapting to a target domain described by text only. Fig. 9 shows the comparison between our DiFa and StyleGAN-NADA [7]. One can see that adapted images from our DiFa are more consistent with their corresponding source images, thereby inheriting more diversity from the pre-trained generator. For example, when performing adaption from FFHQ to a target domain described by "*A sketch with black pencils*", all eyes in StyleGAN-NADA results look to the left, which is inconsistent with the original eyes in source domain. More visualizations and the implementation details are given in the *Suppl*.

## 5 Discussion

In this paper, we presented DiFa to address the diverse generation and faithful adaptation issues for one-shot generative domain adaption. In particular, DiFa leverages the difference between the CLIP embedding of the reference image and the embedding of source domain to guide the global-level adaption. To faithfully acquire local-level domain-specific characteristics, we introduce the attentive style loss to align each intermediate token of adapted images with its closest token of the reference image. For highly diverse generation, the selective cross-domain consistency loss is proposed to select and retain the domain-sharing attributes in $\mathcal{W}+$ space. Both qualitative and quantitative experiments show the superiority of our DiFa against state-of-the-arts under a wide range of settings, especially for the cases of large domain gap. Furthermore, our DiFa can be easily extended to zero-shot generative domain adaption with compelling results.

**Limitations.** When there are few domain-sharing attributes between source and target domains, *e.g.*, Church $\rightarrow$ Tiger, our DiFa cannot produce highly diverse images. Fortunately, this issue may be

largely alleviated by adaptively inheriting the prior knowledge from large-scale generators [20,23,26], which are pre-trained on sufficient variety of source domains.

**Broader Impact.** Transferring a pre-trained generator with very limited data plays a crucial role in academia and industry. More specifically, our DiFa provides insights on tasks in computer vision, *e.g.*, data augmentation and few-shot adaption. Meanwhile, our DiFa also makes AI more accessible to the public. On the one hand, users could leverage our method to create the artworks with any desired styles, even without adequate computing and data resources. On the other hand, our work may bring potential concerns on the probability of producing fake images. For example, someone may use our DiFa to spoof other people's portraits, to synthesize deceptive interactions, or even to impersonate public figures to influence political processes. Albeit there are a few potential negative impacts, we believe that they could be well addressed with the development of DeepFake detection and proper protocols. In particular, we could verify the authenticity, integrality, and source of images by adding digital watermarks or signatures. Also, we may employ DeepFake detection technique to analyze the images without digital signatures. Furthermore, our community should help the government to improve corresponding laws and regulations to avoid the abuse of image generation.

## Acknowledgments and Disclosure of Funding

This work was supported in part by the National Key R&D Program of China under Grant No.s 2020AAA0104500, 2021ZD0112100, and the National Natural Science Foundation of China (NSFC) under Grant No. 62006064.

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
