# Appendix

## A  Outline

Our code is available at . In this appendix, we begin to show more visualization results for zero-shot and one-shot generative domain adaption in Sec. B. Additionally, Sec. C presents implementation details under the zero-shot setting. Furthermore, we conduct more ablation studies and comparison experiments in Sec. D and Sec. E respectively. Finally, we elaborate the user study in Sec. F.

## B  More Visualizations

We provide more qualitative results from a wide range of source and target domains. Fig. 20 shows the results converted from the generator pre-trained on FFHQ. Fig. 21 shows the results converted from the generator pre-trained on LSUN CAR. Fig. 22 shows the results converted from the generator pre-trained on LSUN CHURCH. Fig. 23 shows the results converted from the generator pre-trained on AFHQ-Dog. Fig. 15 shows the results of zero-shot generative domain adaption.

## C  Zero-shot Generative Domain Adaption

Compared to one-shot generative domain adaption, we remove the attentive style loss $\mathcal{L}_{local}$ and modify the global-level adaption loss $\mathcal{L}_{global}$ under the zero-shot setting. Specifically, we compute the domain-gap direction $\Delta \boldsymbol{v}_{dom\_text}$ between the CLIP-space embedding $\boldsymbol{v}_{tar\_text}$ of the given text $T_{tar}$ and the text-based embedding $\boldsymbol{v}_{src\_text}$ of source domain:

$$\Delta \boldsymbol{v}_{dom\_text} = \boldsymbol{v}_{tar\_text} - \boldsymbol{v}_{src\_text}, \tag{9}$$

where $\boldsymbol{v}_{tar\_text} = E_T(T_{tar})$ denotes the embedding of target text $T_{tar}$, and $\boldsymbol{v}_{src\_text} = \mathbb{E}_{t_i \in \mathcal{X}_T}[E_T(t_i)]$ indicates the mean embedding of $N_T$ words $\mathcal{X}_T = \{t_i\}_{i=1}^{N_T}$ closest to the mean source image embedding $\boldsymbol{v}_{\bar{A}}$. $E_T$ is the CLIP text encoder. Together with the sample-shift direction $\Delta \boldsymbol{v}_{samp}$ computed by Eq. 2, the text-based global-level adaption loss is defined as:

$$\mathcal{L}_{global\_text} = 1 - \frac{\Delta \boldsymbol{v}_{samp} \cdot \Delta \boldsymbol{v}_{dom\_text}}{|\Delta \boldsymbol{v}_{samp}||\Delta \boldsymbol{v}_{dom\_text}|}, \tag{10}$$

Consequently, the overall training loss is expressed as:

$$\mathcal{L}_{overall\_text} = \mathcal{L}_{global\_text} + \lambda_{scc}\mathcal{L}_{scc}. \tag{11}$$

Empirically, $\lambda_{scc}$ is set to four, and we choose $N_T = 50$ closest words from the dictionary[†] of CLIP. Table 4 shows the chosen words from different source domains.

## D  More Ablation Studies

**The Proportion $\alpha$ of Preserved Attributes.**  To explore the effect of hyper-parameter $\alpha$ in Eq. 6, we conduct experiments with $\alpha$ linearly decreasing from one to zero in Fig. 11. When decreasing the value of $\alpha$, the synthesized images become more similar to the reference image in the domain-specific attributes (*e.g.*, slim and long face) while having less diversity. Noticeably, when $\alpha$ is less than 0.5 or greater than 0.7, the adapted generator fails to retain enough domain-sharing attributes (*e.g.*, gender and hair length) or acquire domain-specific attributes (*e.g.*, slim and long face). Thus, we set $\alpha$=0.5~0.7 in our experiments. Notably, as show in Fig. 14, a larger $\alpha$ enhances the preservation of face identities for FFHQ source domain.

In Fig. 12, we also conduct additional ablation studies on the choice of $\alpha$ when the source and target domains are quite dissimilar (*i.e.*, church→tiger). Since there are few domain-sharing attributes between church and tiger domains, the selective cross-domain consistency loss favors selecting and retaining the attributes with fewer changes (*e.g.*, shape and pose). Specifically, when the value of $\alpha$ is very large (0.7~1), the synthesized images keep the shape of "church" while acquiring the fur and stripes of "tiger". After linearly decreasing the value of $\alpha$, we observe that the adapted generator produces images with coarser-scale characteristics of the target domain (*e.g.*, the face shape of the tiger). Until $\alpha$ is decreased to zero, all adapted images look very similar to the reference image with little diversity.

---

[†]

**Layer Choice of Attentive Style Loss.** We also investigate the effect of different layer choices on the performance of attentive (AS) style loss. In Fig. 18, all intermediate layers are divide into fine-level (1-2), middle-level (3-6), and coarse-level (7-12). We can observe that fine-level layers only capture fine-grained characteristics (*e.g.*, fur color) of the reference image. Coarse-level layers obtain similar performance with the one w/o AS, because intermediate tokens become more similar to the final CLIP-space embedding as the layers deepen. In contrast, middle-level layers acquire both representative domain styles (*e.g.*, fur color and stripes) and attributes (*e.g.*, mane), hence, we use intermediate tokens from the 4-th layer of CLIP image encoder by default.

**Quantitative Ablation Studies of Proposed Losses.** We have added quantitative ablation studies on the effectiveness of our proposed two losses in Table 6 and 7. From the tables, both selective cross-domain consistency loss and attentive style losses can boost the performance of one-shot domain adaption in terms of KID and FID scores, which is consistent with the qualitative ablation studies.

## E    More Comparison Experiments

**CLIP-based AS vs VGG-based AS.** Fig. 19 shows comparisons between CLIP-based and VGG-based attentive style (AS) loss. As can be seen, VGG-based AS only captures some visual styles (*e.g.*, stripes) from the reference, while CLIP-based AS acquires more representative domain styles (*e.g.*, fur color and stripes) and attributes (*e.g.*, mane), even using one intermediate layer only.

**Our DiFa vs Adversarial Loss Methods.** We present the qualitative comparisons with FSGAN [26] in Fig. 24 and quantitative comparisons with FSGAN [26] and GenDA [36] in Table 8 and 9. As shown in Fig. 24, FSGAN [26] not only suffers from severe mode collapse but also fails to capture domain-specific styles of the reference images. In terms of quantitative results, our DiFa significantly outperforms the methods based on adversarial loss by the KID and FID metrics under the one-shot setting, which is consistent with qualitative results.

**Attentive Style Loss vs Style Mixing** With the aid of intermediate tokens of CLIP model, our attentive style loss directly encourages the model to learn to acquire the target styles. While style mixing acquires the target styles through the obtained latent code of the reference image, and the style heavily relies on the latent code. Usually, it is difficult to faithfully obtain the latent code of the reference image, especially for the images with rare or unseen attributes for the source domain. Therefore, our attentive style loss is more robust than the style mixing trick when dealing with cases involving a large domain gap, *e.g.*, Cat→Tiger in Fig. 17(b)) and using unaligned reference images in Fig. 13.

To quantitatively evaluate the shape discrepancy of faces, we calculate the distances between the landmarks of two different faces. In particular, we use the dlib library[‡] to detect 68 landmarks of the human face and take the Euclidean distance between landmarks of a reference image and a source image as their shape discrepancy. Fig. 16 illustrates the comparison between the style mixing method and our DiFa as the increase of the shape discrepancy. As one can see, the style mixing method synthesizes images with more visible artifacts when increasing the shape discrepancy. In contrast, our DiFa is minimally affected by the shape discrepancy and keeps producing images with high quality and diversity.

**One-stage vs Two-stage Methods.** In Fig. 17, we present the results of a two-stage method (Mind The Gap [42]). One can see that the two-stage method ignores some domain-specific attributes (*e.g.*, red lips in row 1 of Fig. 17(a), manes and stripes in row 1 of Fig. 17(b)), even using the style mixing trick during inference. Specifically, the two-stage method finds the corresponding image in source domain of the reference image and treats its CLIP embedding as source domain embedding. As shown in Fig. 17, the found corresponding image contains some domain-specific attributes of the reference image (*e.g.*, glaze color and red lips in Fig. 17(a)). And the domain gap on these attributes is negligible, thereby ignoring these domain-specific attributes during adaption. Albeit the two-stage method tries to re-acquire ignored domain-specific attributes using style mixing, it still fails to acquire some of them (*e.g.*, red lips in row 1 of Fig. 17) or misunderstands some attributes (e.g., mistake the green hat as green hair).

---

[‡] http://dlib.net/face_landmark_detection.py.html

Table 4: **The 50 words closest to the mean embedding $v_{\bar{A}}$ of source images on different source domains.**

| Source Domain | Chosen Words |
|---|---|
| FFHQ | "person", "headshot", "participant", "face", "closeup", "filmmaker", "author", "pknot", "contestant", "associate", "individu", "volunteer", "michele", "artist", "director", "researcher", "cropped", "lookalike", "mozam", "ml", "portrait", "organizer", "kaj", "coordinator", "appearance", "psychologist", "jha", "pupils", "subject", "entrata", "newprofile", "guterres", "staffer", "diem", "cosmetic", "viewer", "assistant", "writer", "practitioner", "adolescent", "white", "elling", "nikk", "addic", "onnell", "customer", "client", "simone", "greener", "candidate" |
| AFHQ-Cat | "burmese", "feline", "cat", "tabby", "cathedr", "gata", "gato", "alcat", "catt", "pupils", "wildcat", "bengal", "tuna", "artemis", "feral", "persian", "meow", "figaro", "packet", "java", "cappuccino", "tora", "alley", "catal", "chipped", "kitty", "cathar", "miaw", "pye", "chattanoo", "katz", "sniff", "kerswednesday", "peuge", "categor", "nak", "mae", "catalo", "scratch", "tabern", "plume", "striped", "chat", "catsofinstagram", "cajun", "meredith", "offee", "sylvester", "popart", "pling" |
| AFHQ-Dog | "adog", "dog", "canine", "adoptable", "doggie", "doggy", "mutt", "pupp", "doggo", "terrier", "pup", "cajun", "pooch", "dogday", "maverick", "dawg", "watchdog", "lostdog", "peuge", "woof", "skye", "tucker", "sampson", "detect", "dug", "kodi", "embark", "renegade", "puppy", "wrangler", "hula", "ruff", "sabre", "zeus", "dharma", "wag", "cooper", "brownie", "aviator", "kita", "bud", "cigar", "shepherd", "chaser", "dixie", "taro", "scotch", "duke", "tobi", "bullet" |
| LSUN CAR | "car", "ecar", "vehicle", "automobile", "automotive", "autonews", "icar", "hatchback", "incar", "auto", "saab", "sedan", "lowered", "classiccar", "cars", "nissan", "stance", "citroen", "supercharged", "tuned", "facelift", "volvo", "convertible", "forza", "skoda", "civic", "sportscar", "oem", "opel", "mazda", "valet", "extravag", "merc", "].", "parked", "chevrolet", "suv", "coupe", "slammed", "xf", "toyota", "gtx", "bmw", "corsa", "tdi", "taxi", "amg", "detailing", "peugeot", "spoiler" |
| LSUN CHURCH | "cathedral", "church", "churches", "basilica", "chapel", "anglican", "lutheran", "diocese", "dral", "presbyterian", "apse", "friars", "st", "cathol", "methodist", "abbey", "baptist", "synagogue", "conduc", "argu", "assumption", "jesu", "congregation", "priory", "nave", "episcopal", "halle", "exterior", "cst", "sedly", "echel", "mably", "gonzaga", "nd", "protestant", "thex", "monastery", "cour", "bishops", "mons", "minster", "tor", "sacrific", "shul", "heritag", "sque", "restoration", "wul", "spires", "notre" |

Table 5: **Statistics of participates in user study.**

| Factors | Statistics |
|---|---|
| Gender | Male: 53.3%, Female: 47.7% |
| Age | $\leq$20: 23.3%, 20~40: 56.7%, $\geq$ 40: 20% |
| Background | CV and CG: 33.3%, Arts: 36.7%, Other% |
| Race | Caucasian: 26.7%, Mongoloid: 33.3%, Negroid: 23.3%, Australoid: 16.7% |

# F  User Study

We perform user study to further compare our DiFa with other approaches, from the perspective of (i) image quality, (ii) style similarity and (iii) attribute consistency. We recruit 30 participates from both universities and industries, whose statistics are shown in Table 5. Particularly, we randomly generate 1,050 samples for each "our DiFa vs another method" comparison. Afterwards, we assign these samples to 30 participates and ask them to complete the survey following the instructions in Fig. 10. Finally, we collect their answers and illustrate the statistics in Table 3.

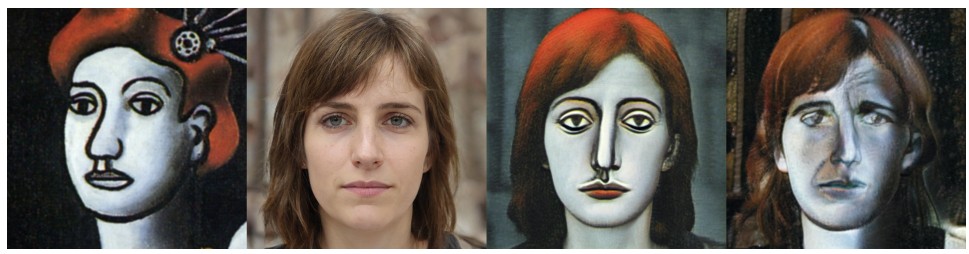

| Reference | Source image | Method 1 | Method 2 |

Between Method 1 & 2 :
1. Which image has higher quality ?
2. Which image has a more similar style to the reference ?
3. Which image has more consistent attributes with source image ?

Figure 10: **Instructions of user study.** A user study sample consists of a reference, a source image, and two adapted images from our DiFa and another method. The participates are asked to answer the above three questions for each sample. Note that two adapted images are randomly permuted to reduce potential position bias.

Table 6: **KID (↓) of ablation studies on selective cross-domain consistency (SCC) and attentive style (AS) losses.** Each result is averaged over 5 training shots and in the form of {mean ± standard error}.

| w/ SCC | w/ AS | Amedeo. | Fernand. | Raphael |
|--------|-------|---------|----------|---------|
| | | 131.03±28.14 | 169.83± 31.52 | 149.19± 55.91 |
| ✓ | | 129.41± 26.17 | 165.73± 31.75 | 119.36± 20.18 |
| ✓ | ✓ | **121.21±24.62** | **159.93±31.39** | **112.72±17.61** |

Table 7: **FID (↓) of ablation studies on selective cross-domain consistency (SCC) and attentive style (AS) losses.** Each result is averaged over 5 training shots and in the form of {mean ± standard error}.

| w/ SCC | w/ AS | Amedeo. | Fernand. | Raphael |
|--------|-------|---------|----------|---------|
| | | 188.44±19.15 | 257.27±19.39 | 186.20±28.60 |
| ✓ | | 187.42±19.32 | 257.18±21.32 | 180.61±15.32 |
| ✓ | ✓ | **187.28±24.45** | **254.68±17.73** | **172.34±10.15** |

Table 8: **KID (↓) comparisons between our DiFa and adversarial loss based methods.** Each result is averaged over 5 training shots and in the form of {mean ± standard error}.

| Models | Amedeo. | Fernand. | Raphael | Sketches |
|--------|---------|----------|---------|----------|
| FSGAN [26] | 299.64± 34.16 | 348.70± 41.27 | 151.79± 35.12 | 227.78±12.71 |
| **Ours** | **121.21± 24.62** | **159.93±31.39** | **112.72±17.61** | **53.24±7.82** |

Table 9: **FID (↓) comparisons between our DiFa and adversarial loss based methods.** Each result is averaged over 5 training shots and in the form of {mean ± standard error}. * indicates that results are from the original paper.

| Models | Amedeo. | Fernand. | Raphael | Sketches |
|--------|---------|----------|---------|----------|
| FSGAN [26] | 288.75±46.76 | 360.45±58.07 | 200.29±78.23 | 166.37±11.32 |
| GenDA* [36] | - | - | - | 87.55 |
| **Ours Ours** | **187.28±24.45** | **254.68±17.73** | **172.34±10.15** | **56.93±5.48** |

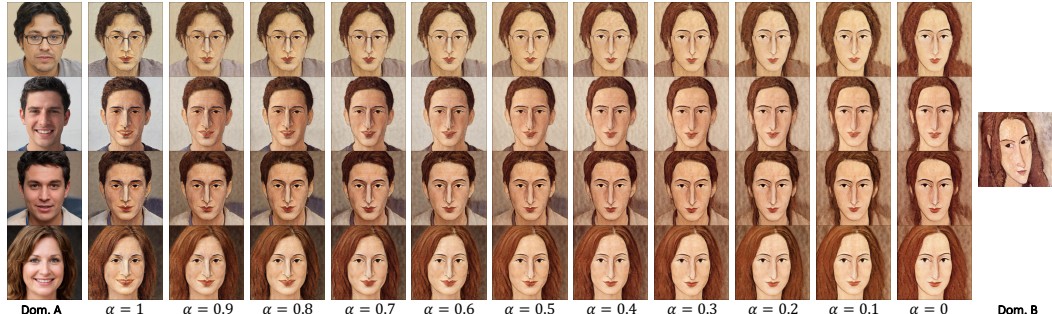

Figure 11: **Ablation studies on the proportion $\alpha$ of preserved attributes.** The first and last column show the source images from domain $A$ and the reference image from domain $B$. The other columns show the results using linearly decreasing $\alpha$ from one to zero, *i.e.*, preserving less attributes.

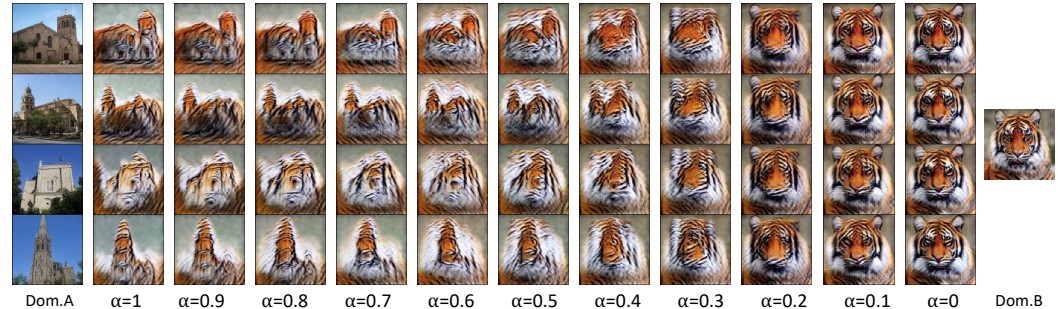

Figure 12: **Ablation studies on the proportion $\alpha$ of preserved attributes for dissimilar domains (church $\rightarrow$ tiger).** The first and last column show the source images from domain $A$ and the reference image from domain $B$. The other columns show the results using linearly decreasing $\alpha$ from one to zero, *i.e.*, preserving less attributes.

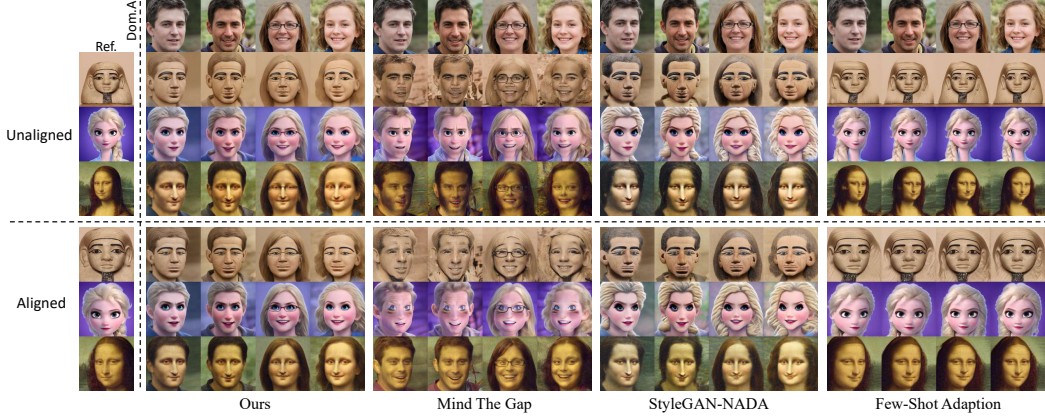

Figure 13: **Qualitative results using unaligned and aligned reference images.** The first row and column show the source images from domain $A$ and the reference image from domain $B$.

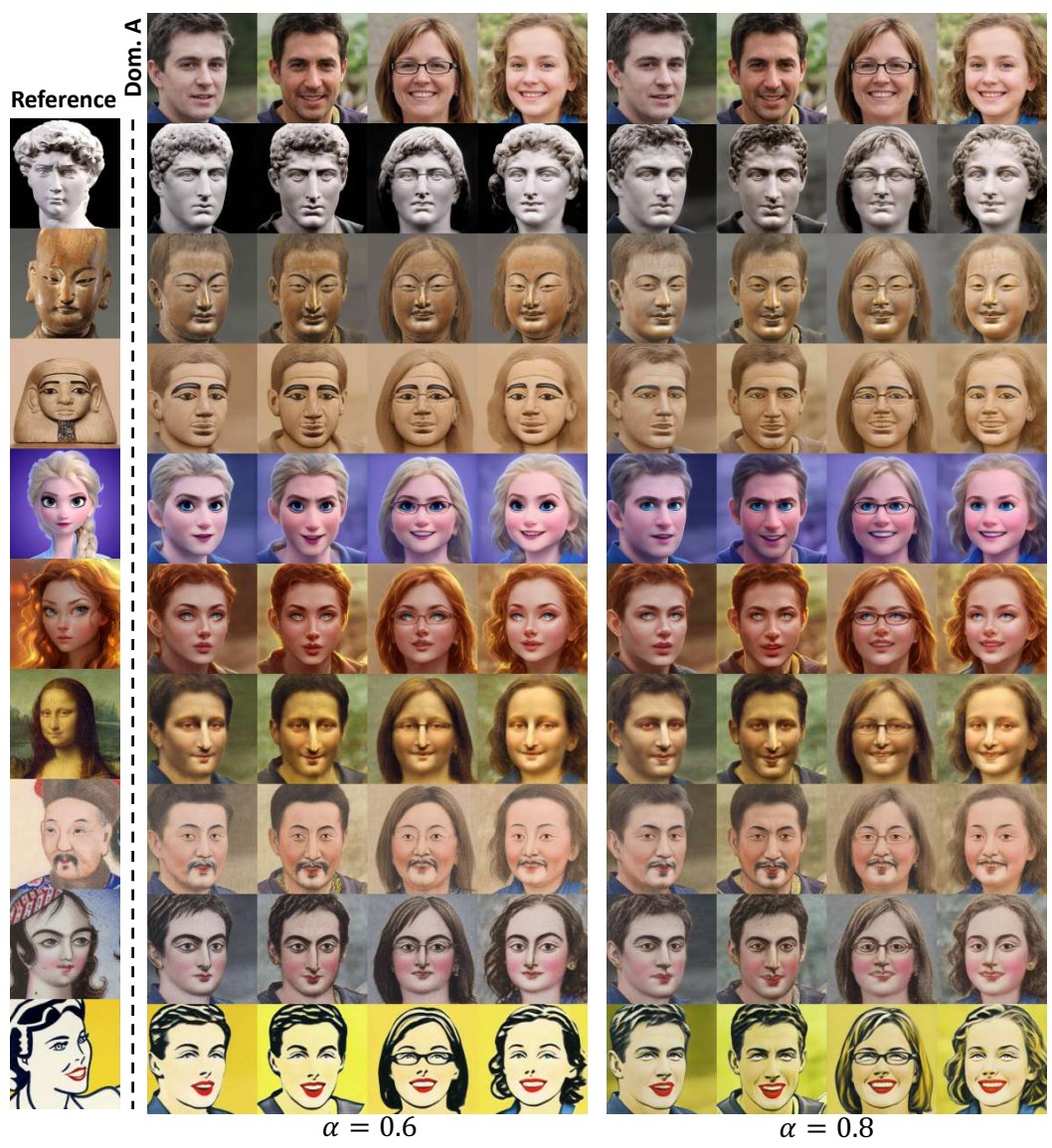

Reference Dom. A

$\alpha = 0.6$        $\alpha = 0.8$

Figure 14: **Qualitative results comparisons** $\alpha$**=0.6 (original) and** $\alpha$**=0.8 using the generator pre-trained on FFHQ**. The first row and first column show source images in domain $A$ and reference images in domain $B$. **Results best seen at 500% zoom.**

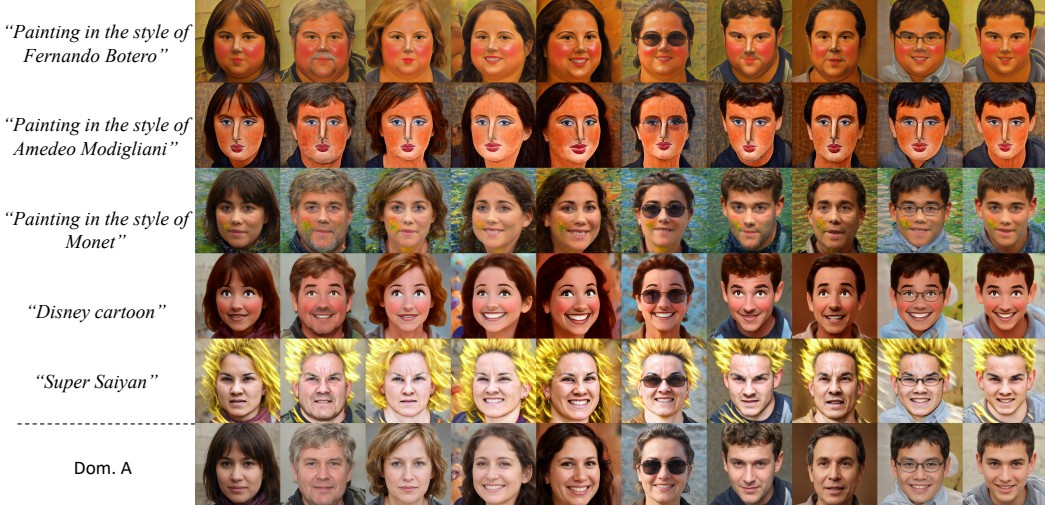

Figure 15: **Qualitative results for zero-shot generative domain adaption.** The last row and first column show the source images from domain $A$ and the descriptions of domain $B$, respectively. **Results best seen at 500% zoom.**

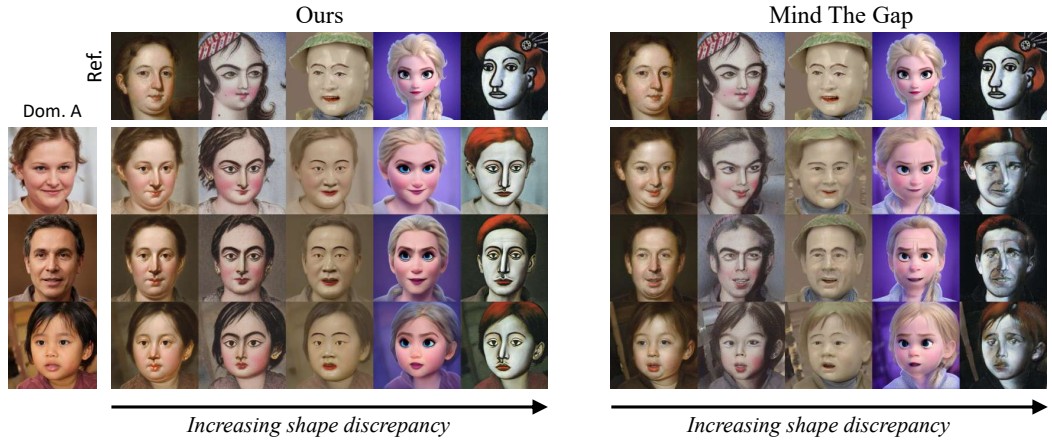

Figure 16: **Qualitative comparisons when increasing the shape discrepancy between source images and the reference image** between our DiFa and Mind The Gap [42]. The first row and first column show reference images in domain $B$ and source images in domain $A$. The shape discrepancy between a reference image and a source image is defined as the L2 normalized distance between their face landmarks. When increasing the shape discrepancy from left to right, our DiFa produces images with high quality and diversity while Mind The Gap produces images with more visible artifacts.

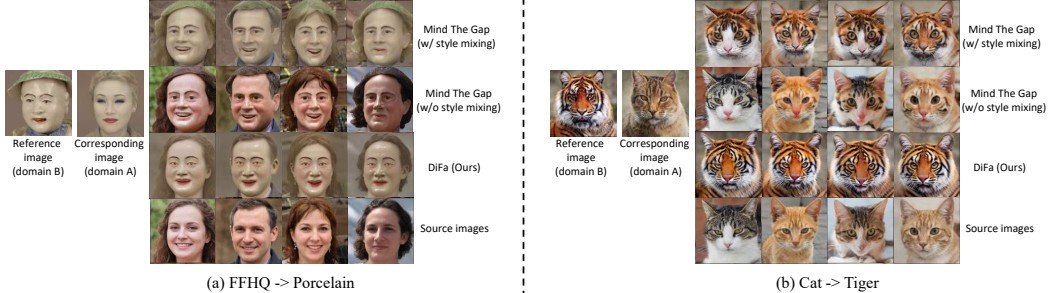

Figure 17: **Qualitative comparisons between attentive loss based (our DiFa) and style mixing based method (Mind The Gap [42]).** For both (a) and (b), the first and second columns show reference images in domain $B$ and their corresponding images in domain $A$ respectively. The first row shows source images in domain $A$.

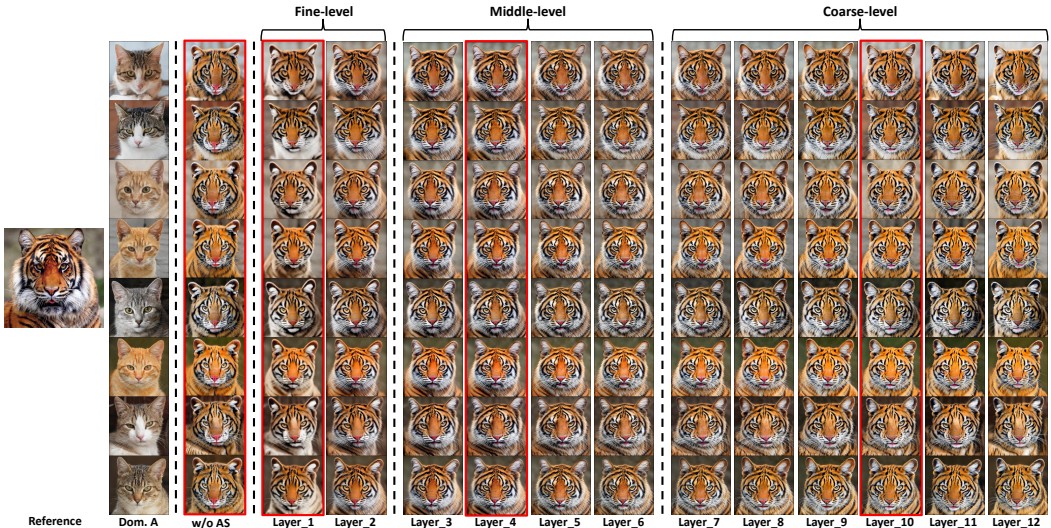

Figure 18: **Ablation studies on the layer choice of attentive style (AS) loss.** The first two column show the reference image from domain $B$ and the source images from domain $A$. The other columns show the results under the different AS configurations, *i.e.*, using the intermediate tokens from different layers of CLIP image encoder. Akin to [10], we divide all layers into fine-level, middle-level, and coarse-level, and then select a layer (in red box) from each stage for better comparison.

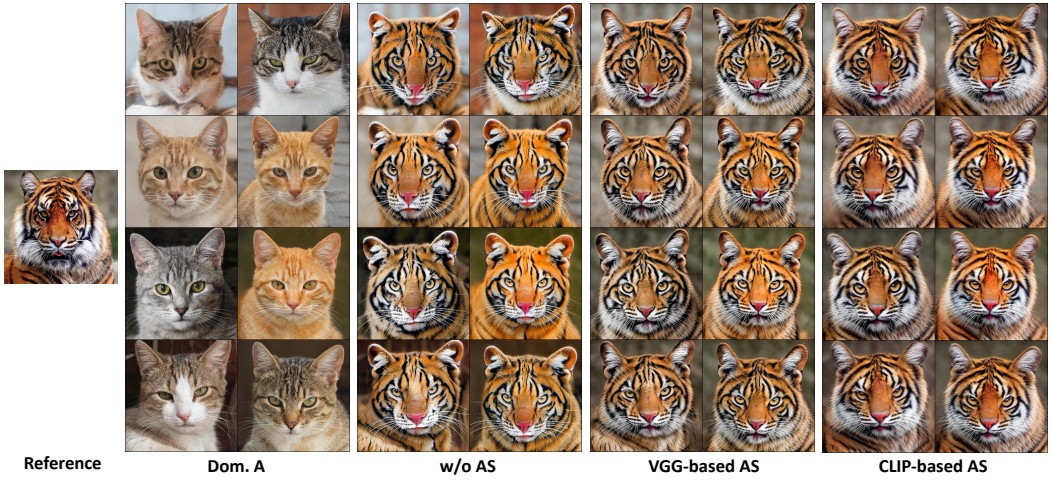

Figure 19: **CLIP-based vs. VGG-based attentive style (AS) loss.** The first two columns show the reference from domain $B$ and the source images from domain $A$ and the reference images from domain $B$, respectively. Akin to [15], VGG-based AS uses all layers of VGG16 but layers 9, 10, 12, and 13. In contrast, our CLIP-based AS only uses the 4-th layer of CLIP image encoder.

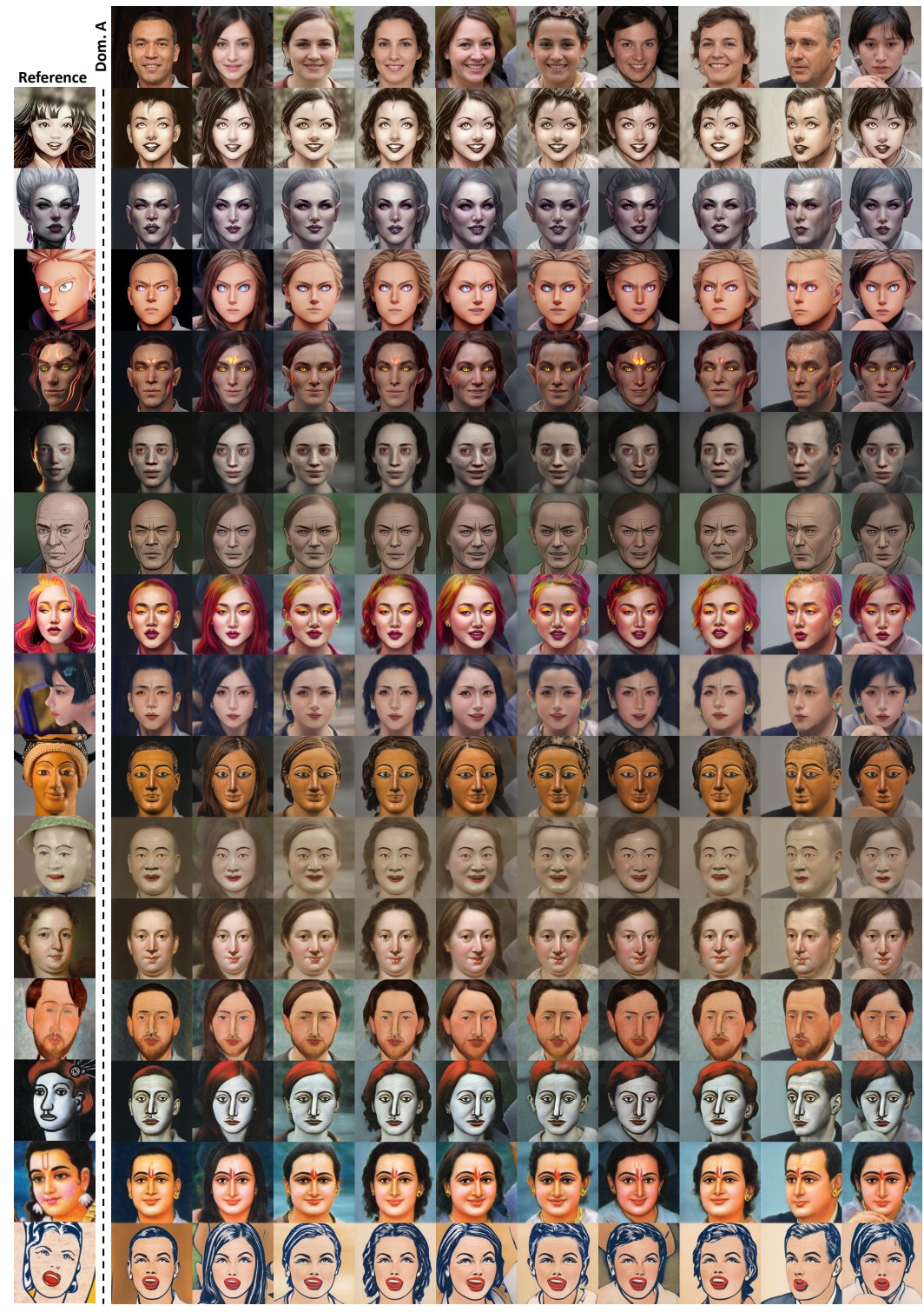

Figure 20: **Qualitative results using the generator pre-trained on FFHQ.** The first row and column show the source images from domain $A$ and the reference images from domain $B$, respectively. **Results best seen at 500% zoom.**

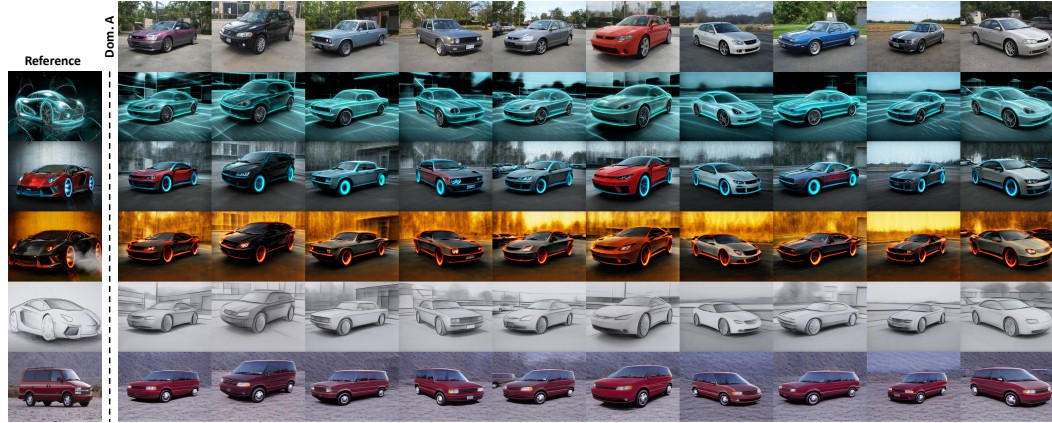

Figure 21: **Qualitative results using the generator pre-trained on LSUN CAR.** The first row and column show the source images from domain $A$ and the reference images from domain $B$, respectively. **Results best seen at 500% zoom.**

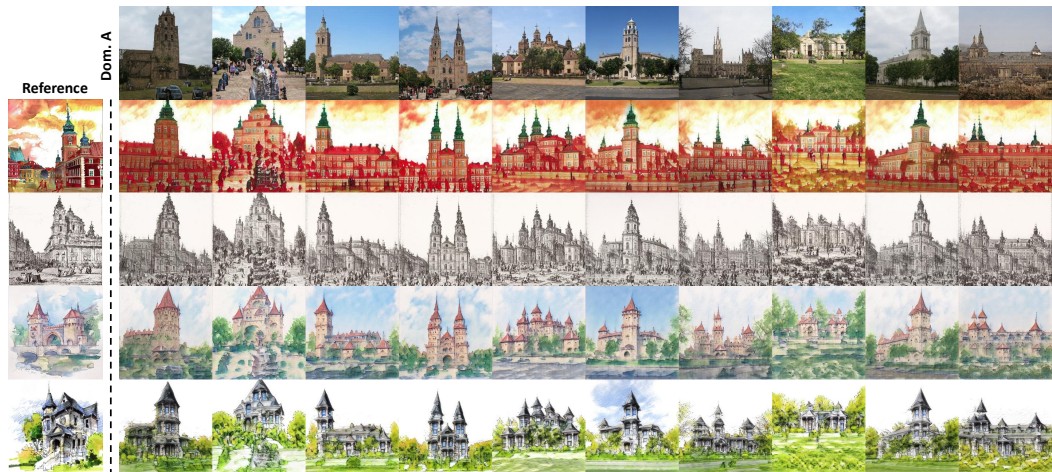

Figure 22: **Qualitative results using the generator pre-trained on LSUN CHURCH.** The first row and column show the source images from domain $A$ and the reference images from domain $B$, respectively. **Results best seen at 500% zoom.**

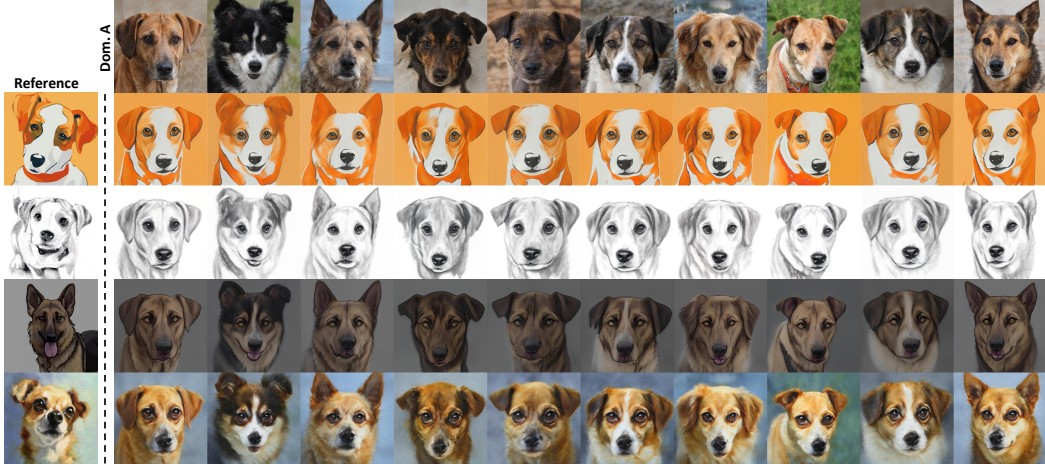

Figure 23: **Qualitative results using the generator pre-trained on AFHQ-Dog.** The first row and column show the source images from domain $A$ and the reference images from domain $B$, respectively. **Results best seen at 500% zoom.**

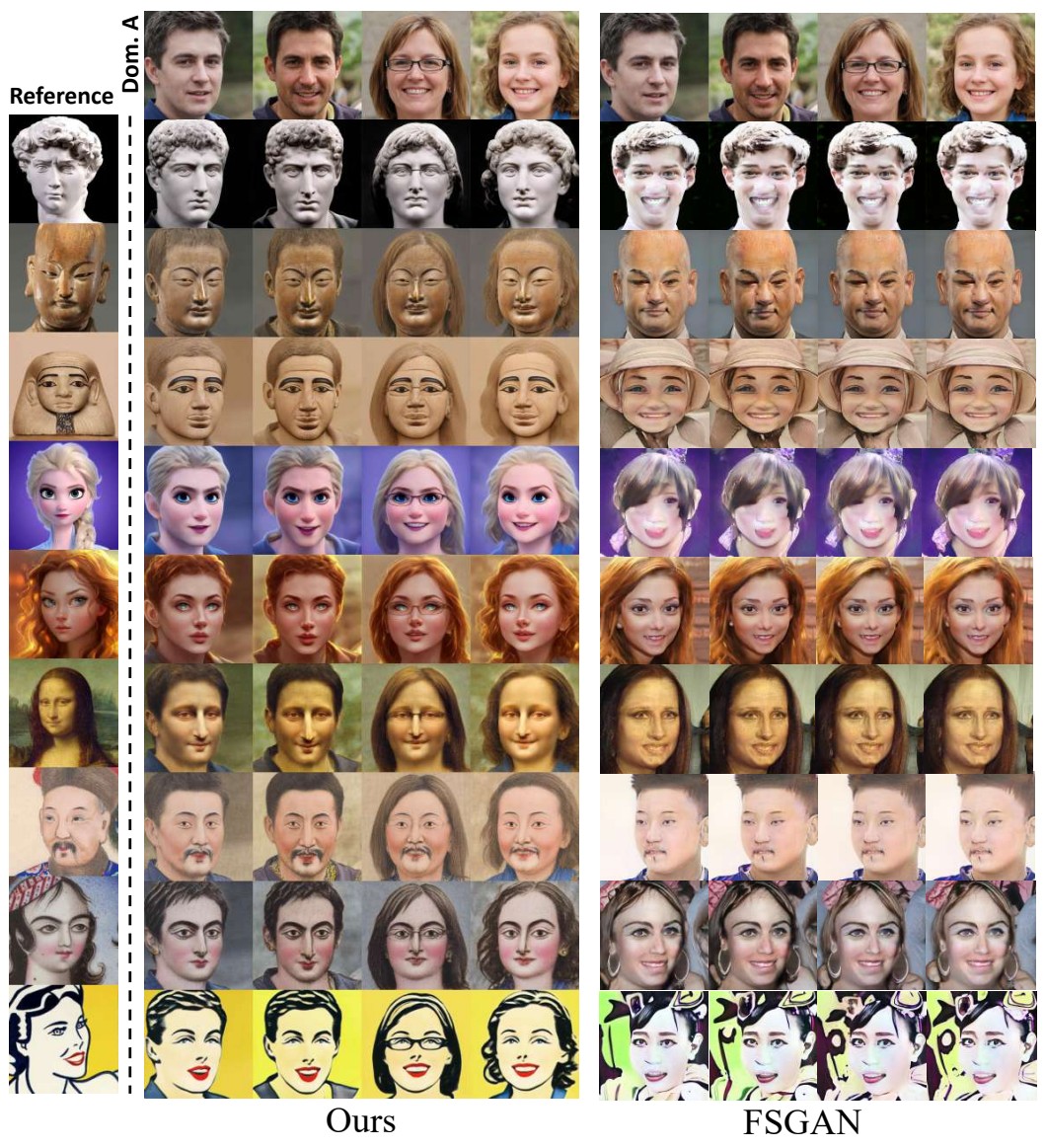

Figure 24: **Qualitative comparisons using the generator pre-trained on FFHQ** [12] between our DiFa, and FSGAN [26]. The first row and first column show source images in domain $A$ and reference images in domain $B$. In contrast, FSGAN suffers from severe mode collapse and fails to obtain domain-specific styles of the reference images. **Results best seen at 500% zoom.**