# OpenReview forum: "Towards Diverse and Faithful One-shot Adaption of Generative Adversarial Networks"
_NeurIPS.cc/2022/Conference — NeurIPS 2022 Accept_

### Official Review · Reviewer_kovL · 2022-07-10

**Rating:** 7
**Confidence:** 3
**Soundness:** 3 good
**Presentation:** 3 good
**Contribution:** 2 fair

**Summary:**

The paper proposes a CLIP-based method for generative domain adaptation. The method starts with a (StyleGAN2) generator that is pretrained on a source domain and uses it to train a new version of the generator that can produce images from a target domain, from which only a single reference image is available. During training, a global loss is used to reduce the distance between the CLIP embeddings from the two domains. Furthermore, an attentive style loss is used to capture the local style of the target domain, by aligning intermediate tokens of the CLIP encoder for the generated images and the reference image. Finally, a selective cross-domain consistency loss, that operates in the latent space obtained from an inversion model, is used to preserve the diversity of attributes shared by both domains. The proposed method can also be adapted for use in a zero-shot setting and is compared to other GAN-based domain adaptation methods

**Questions:**

- Since the α hyperparameter is proportional to the number of shared attributes assumed between domains, how does its choice vary when the two domains are quite dissimilar.
- The paper claims that FSA has better scores yet suffers from extreme mode collapse. Indeed from visual inspection based on the provided samples. However, FID in theory should correlate not only with perceptual quality but also with diversity. Do the authors have any hypothesis about why the FID of FSA is better in two experiments?
- The paper mentions that for dissimilar domains (church -> tiger) the end result lacks diversity.  What do such results look like? Is it close to the reference image?

**Limitations:**

Regarding the limitations of the method, the authors mention that for dissimilar domains (church -> tiger) the end result lacks diversity but this would be better understood by the reader if some examples were included in the paper or supplementary material.

The authors have included a discussion to raise awareness about the potential negative social impact of their work.

**Strengths And Weaknesses:**

## Strengths:
- The method is well motivated and the paper is clearly written.
- The additional ablation study in figure 10 of the supplementary material gives the reader a good insight into the effect of the α parameter.
- Based on visual assessment and the quantitative evaluation of the method it seems to outperform other similar GAN-based domain adaptation methods.

## Weaknesses:
- The contributions of the paper are not immediately clear from the introduction. The authors should make clear that the novelty in the proposed approach is in the attentive style and the selective cross-domain consistency losses.
- The ablation study is based entirely on a visual assessment of one example of style transfer. The authors should ideally also provide the FID for each version of the model in the ablation study.
- Some points made in the paper require clarification (see Questions section of the review)

---

> ### Author Response · Authors · 2022-08-02
> **Response to Reviewer kovL**
>
> Thanks for your constructive comments.
>
> **The contributions of the paper are not immediately clear from the introduction.**
>
> Thanks for your kind suggestion. We have added a summary of our contributions in our revision as follows:
> > Overall, our contributions are summarized as follows:
> (1) a novel method DiFa for one-shot generative domain adaption,
> (2) the attentive style and selective cross-domain consistency losses for diverse generation and faithful adaption,
> (3) the extension to the zero-shot setting with appealing results.
>
> ------------
>
> **The FID for each version of the model in the ablation study.**
>
> We have added quantitative ablation studies on the effectiveness of our proposed two losses in the tables below.
> From the tables, both selective cross-domain consistency loss and attentive style losses can boost the performance of one-shot domain adaption in terms of KID and FID scores, which is consistent with the qualitative ablation studies in Fig. 7 of the paper.
>
> Table 5. KID ($\downarrow$) of ablation studies on selective cross-domain consistency (SCC) and attentive style (AS) losses. Each result is averaged over 5 training shots and in the form of \{mean $\pm$ standard error\}.
> | w/ SCC | w/ AS | Amedeo.              | Fernand.             | Raphael              |
> | ------ | ----- | -------------------- | -------------------- | -------------------- |
> | No     | No    | 131.03$\pm$28.14     | 169.83$\pm$31.52     | 149.19$\pm$55.91     |
> | Yes    | No    | 129.41$\pm$26.17     | 165.73$\pm$31.75     | 119.36$\pm$20.18     |
> | Yes    | Yes   | **121.21$\pm$24.62** | **159.93$\pm$31.39** | **112.72$\pm$17.61** |
>
>
> Table 6. FID ($\downarrow$) of ablation studies on selective cross-domain consistency (SCC) and attentive style (AS) losses. Each result is averaged over 5 training shots and in the form of \{mean $\pm$ standard error\}.
> | w/ SCC | w/ AS | Amedeo.              | Fernand.             | Raphael              |
> | ------ | ----- | -------------------- | -------------------- | -------------------- |
> | No     | No    | 188.44$\pm$19.15     | 257.27$\pm$19.39     | 186.20$\pm$ 28.60    |
> | Yes    | No    | 187.42$\pm$19.32     | 257.18$\pm$21.32     | 180.61$\pm$15.32     |
> | Yes    | Yes   | **187.28$\pm$24.45** | **254.68$\pm$17.73** | **172.34$\pm$10.15** |
>
> We have also updated the results in Table 5 and 6 of the revised supplementary materials.
>
> **How does the choice of the hyperparameter $\alpha$ vary when the two domains are quite dissimilar.**
>
> We have conducted additional ablation studies on the choice of $\alpha$ when the source and target domains are quite dissimilar (*i.e.* church $\to$ tiger), and qualitative results are shown in Fig. 19 of the revised supplementary materials.
> Since there are few domain-sharing attributes between church and tiger domains, the selective cross-domain consistency loss favors selecting and retaining the attributes with fewer changes (*e.g.*, shape and pose).
> Specifically, when the value of $\alpha$ is very large (0.7$\sim$1), the synthesized images keep the shape of "church" while acquiring the fur and stripes of  "tiger".
> After linearly decreasing the value of $\alpha$, we observe that the adapted generator produces images with coarser-scale characteristics of the target domain (e.g., the face shape of the tiger).
> Until $\alpha$ is decreased to zero, all adapted images look very similar to the reference image with little diversity.
>
> **The hypothesis about why the FID of FSA is better in two experiments.**
>
> One possible explanation is that these two small datasets have different data biases with FFHQ, *e.g.* gender bias.
> Specifically, 8/10 images in the Amedeo Modigliani dataset and 9/10 images in the Fernand Leger dataset are female.
> Due to our DiFa acquiring the diversity from the original generator which is trained on FFHQ, it generates male and female adapted images with similar probability.
> In contrast, for FSA,  the adapted images are all similar to the reference image.
> When comparing on the above two datasets, FSA tends to generate images that have similar gender distribution to the validation dataset, thus achieving better FID results.
> For the Raphael dataset, which has 5/10 images that are female, our DiFa achieves better FID results.
>
> **The paper mentions that for dissimilar domains (church -> tiger) the end result lacks diversity. What do such results look like?**
>
> We have added the results of  "church $\to$ tiger'' domain adaption in Fig. 19 of the revised supplementary materials.
> Since there are few domain-sharing attributes between church and tiger domains, we use a low-value $\alpha$ (0$\sim$0.2) for faithful adaption. From the figure we can see that, the synthesized images all look very similar to the reference image and lack diversity.

---

> > ### Comment · Reviewer_kovL · 2022-08-09
> > **Final thoughts**
> >
> > I would like to thank the authors for their response. After the additional results and clarifications I would be happy to increase my rating to an accept (7).

---

> > > ### Author Response · Authors · 2022-08-09
> > > **Response to Reviewer kovL**
> > >
> > > Thanks for your valuable suggestions and positive feedback on our work! They are very helpful for us to improve the paper.

---

> > ### Author Response · Authors · 2022-08-09
> > **Re: Response to Reviewer kovL**
> >
> > Thanks again for your appreciation of our work.  But we find that your rating on our work is still 6. Could you consider changing the rating before ddl?

---

### Official Review · Reviewer_Pk6r · 2022-07-11

**Rating:** 6
**Confidence:** 5
**Soundness:** 3 good
**Presentation:** 3 good
**Contribution:** 3 good

**Summary:**

The paper studies how to transfer a pre-trained generator from source domain to target domain with only one image available. The main challenges of one-shot adaptation of GANs are (i) generating diverse images and (ii) acquiring domain-specific attributes and styles for target domain. This paper proposes DiFa for diverse generation and faithful adaptation. The key ideas are: (i) aligning the CLIP embedding of target and source images at the global-level and aligning the intermediate token of adapted and reference image with an attentive style loss at the local-level; (ii) introducing selective cross-domain consistency loss to facilitate diverse generation. Experiments results demonstrate the efficiency of the proposed techniques.

**Questions:**

- The authors claim that the local attributes and visual styles are usually ignored in global-level CLIP adaptation, then why the local level adaptation, which extract the intermediate tokens of reference and adapted image from CLIP image encoder, can faithfully capture the local-level domain-specific characteristics? Is it possible that the domain-specific characteristics obtained from local-level adaptation are not precise?
- Since both of style-mixing trick and the proposed attentive style loss can capture domain-specific styles. What is the main difference between them? Style-mixing methods may lead to visible artifacts when there is a significant shape discrepancy, is there any quantitative evaluation of such shape discrepancy? As the eyeglasses of the third image in the third-to-last row of Figure 5 (a) disappeared, does the eyeglasses attribute belongs to shape discrepancy?
- The authors claim the two-stage methods find the image corresponding to target image in source domain and treat its CLIP-space embedding as source domain embedding, which may cause the domain gap direction to ignore domain-specific attributes. Is there any experimental or theoretical evidence? In my opinion, such source domain embedding may contain different attributes with adapted images, which may facilitate the diversity.
- How does the proposed method perform when compared to adversarial loss based methods [1] [2] [3]?
- Line 155, “as shown in Fig.4”, should Fig.4 here be Fig.2?

[1] "Few-shot adaptation of generative adversarial networks." arXiv preprint arXiv:2010.11943 (2020).
[2] "One-shot generative domain adaptation." arXiv preprint arXiv:2111.09876 (2021).
[3] "A Closer Look at Few-shot Image Generation." Proceedings of the IEEE/CVF Conference on Computer Vision and Pattern Recognition. 2022.

**Limitations:**

yes

**Strengths And Weaknesses:**

- The technical ideas of this paper are simple and easy-to-follow. It is doable to reproduce the experiments results;
- Strong empirical performance shows the significant advantage of the proposed method on one-shot adaptation of GANs, as well as compelling results when extended to latent space editing and zero-shot domain adaptation of GANs;

---

> ### Author Response · Authors · 2022-08-02
> **Response to Reviewer Pk6r - Part 1**
>
> Thanks for your constructive comments.
>
> **Why the local level adaptation can faithfully capture the local-level domain-specific characteristics? Is it possible that the domain-specific characteristics obtained from local-level adaptation are not precise?**
>
> Many methods such as StyleGAN-NADA and Mind the Gap have shown the superiority of embeddings of CLIP models to capture the target domain-specific styles from the reference images.
> Following existing style transfer methods [5], we assume that the intermediate features (*e.g.* tokens) of CLIP models can also capture the domain-specific styles.
> Each intermediate token of CLIP model corresponds to a specific patch of the reference/adapted image and thus can be used to represent the local attributes and visual styles (*e.g.*, mane and stripes of the tiger in Fig. 12 of the supplementary materials).
> Besides, to alleviate the pose discrepancy and attentively capture local-level characteristics, every patch token from the adapted images is encouraged to approach the patch token with the highest cosine similarity from the reference image.
> With these settings, our attentive style loss can acquire more precise local-level domain-specific characteristics than using global-level adaption only.
> For example, as shown in Fig. 12 of the supplementary materials, the adapted generator using CLIP-based attentive style loss manages to capture the mane and stripes attributes of a tiger, verifying the effectiveness of our proposed local-level adaption method.
>
> **The main difference between style-mixing trick and the proposed attentive style loss.**
>
> With the aid of intermediate tokens of CLIP model, our attentive style loss directly encourages the model to learn to acquire the target styles. While style mixing acquires the target styles through the obtained latent code of the reference image, and the style heavily relies on the latent code. Usually, it is difficult to faithfully obtain the latent code of the reference image, especially for the images with rare or unseen attributes for the source domain. Therefore, our attentive style loss is more robust than the style mixing trick when dealing with cases involving a large domain gap (*e.g.* Cat $\to$ Tiger in Fig. 20(b)). As shown in Fig. 20 in supplementary materials, with the attentive style loss, our DiFa faithfully acquires the glaze color and red lips of "porcelain" (*i.e.*, Fig. 20 (a)), and the fur color and manes of "tiger" under two large-domain-gap settings. In contrast, Mind The Gap (using the style-mixing trick) mistakes the color of the hat (*i.e.*, green) as the color of hair under the "FFHQ $\to$ Porcelain" setting and ignores some domain-specific attributes of tiger, *i.e.*, manes and stripes, under the "Cat $\to$ Tiger" setting.
>
> **Style-mixing methods may lead to visible artifacts when there is a significant shape discrepancy, is there any quantitative evaluation of such shape discrepancy?**
>
> One feasible metric to quantitatively evaluate the shape discrepancy of faces is calculating the distances between the landmarks of two different faces.
> In particular, we use the dlib library to detect 68 landmarks of the human face and take the L2 distance between landmarks of a reference image and a source image as their shape discrepancy.
> Fig. 23 of the revised supplementary materials illustrates the comparison between the style mixing method and our DiFa as the increase of the shape discrepancy.
> As one can see, the style mixing method synthesizes images with more visible artifacts when increasing the shape discrepancy.
> In contrast, our DiFa is minimally affected by the shape discrepancy and keeps producing images with high quality and diversity.
>
>
> **As the eyeglasses of the third image in the third-to-last row of Figure 5 (a) disappeared, does the eyeglasses attribute belongs to shape discrepancy?**
>
> The eyeglasses attribute should not belong to shape discrepancy.
> One possible explanation is that we adopt the same hyper-parameters for any domains instead of adopting specific ones for face domain.
> In Fig. 21 of the revised supplementary materials, we provide additional results with a larger $\alpha$ (0.6 --> 0.8) and show that the eyeglasses of the third image in the third-to-last row become more visible.

---

> ### Author Response · Authors · 2022-08-02
> **Response to Reviewer Pk6r - Part 2**
>
> **Which may cause the domain gap direction to ignore domain-specific attributes in two-stage methods?**
>
> In Fig. 20 of the revised supplementary materials, we present the results of a two-stage method (Mind The Gap [4]).
> One can see that the two-stage method ignores some domain-specific attributes (*e.g.*, red lips in row 1 of Fig. 20 (a), manes and stripes in row 1 of Fig. 20 (b)), even using the style mixing trick during inference.
> Specifically, the two-stage method finds the corresponding image in source domain of the reference image and treats its CLIP embedding as source domain embedding.
> As shown in Fig. 20, the found corresponding image contains some domain-specific attributes of the reference image (*e.g.*, glaze color and red lips in Fig. 20 (a)).
> And the domain gap on these attributes is negligible, thereby ignoring these domain-specific attributes during adaption.
> Albeit the two-stage method tries to re-acquire ignored domain-specific attributes using style mixing, it still fails to acquire some of them (*e.g.*, red lips in row 1 of Fig. 20) or misunderstands some attributes (*e.g.*, mistake the green hat as green hair).
>
> **Comparison with adversarial loss based methods [1] [2] [3].**
>
> We present the qualitative comparisons with FSGAN [1] in Fig. 22 of the supplementary materials and quantitative comparisons with FSGAN [1] and GenDA [2] in Table 7 and 8 of the supplementary materials (or below).
> As shown in Fig. 22, FSGAN [1] not only suffers from severe mode collapse but also fails to capture domain-specific styles of the reference images.
> In terms of quantitative results, our DiFa significantly outperforms the methods based on adversarial loss by the KID and FID metrics under the one-shot setting, which is consistent with qualitative results.
>
> Table 7. KID ( ↓ ) comparisons between our DiFa and adversarial loss based methods.  Each result is averaged over 5 training shots and in the form of { mean ± standard error } .
> | Model  | Amedeo. | Fernado. | Raphael | Sketches |
> | ----- | ------- | ------- | ------- | -------- |
> | FSGAN [1] | 299.64$\pm$34.16 | 348.70$\pm$41.27 | 151.79$\pm$35.12 | 227.78$\pm$12.71 |
> | **Ours** | **121.21$\pm$24.62** | **159.93$\pm$31.39** | **112.72$\pm$17.61** | **53.24$\pm$7.82** |
>
>
> Table 8. FID ( ↓ ) comparisons between our DiFa and adversarial loss based methods. Each result is averaged over 5 training shots and in the form of { mean ± standard error } . * indicates that results are from the original paper.
> | Model    | Amedeo.              | Fernado              | Raphael              | Sketches           |
> | -------- | -------------------- | -------------------- | -------------------- | ------------------ |
> | FSGAN [1]    | 288.75$\pm$46.76     | 360.45$\pm$58.07     | 200.29$\pm$78.23     | 166.37$\pm$11.32   |
> | GenDA [2]*   | -                    | -                    | -                    | 87.55              |
> | **Ours** | **187.28$\pm$24.45** | **254.68$\pm$17.73** | **172.34$\pm$10.15** | **56.93$\pm$5.48** |
>
> -------------
>
> **Line 155, “as shown in Fig.4”, should Fig.4 here be Fig.2?**
>
> Thanks for pointing it out. Fig. 4 in line 155 should be Fig. 3, and we have fixed it in our revision.
>
> -------------
>
> [1] "Few-shot adaptation of generative adversarial networks." arXiv preprint arXiv:2010.11943 (2020).
>
> [2] "One-shot generative domain adaptation." arXiv preprint arXiv:2111.09876 (2021).
>
> [3] "A Closer Look at Few-shot Image Generation." Proceedings of the IEEE/CVF Conference on Computer Vision and Pattern Recognition. 2022.
>
> [4] "Mind the gap: Domain gap control for single shot domain adaptation for generative adversarial networks." arXiv preprint arXiv:2110.08398, (2021).
>
> [5] "Style transfer by relaxed optimal transport and self-similarity." Proceedings of the IEEE/CVF Conference on Computer Vision and Pattern Recognition. 2019.

---

> ### Author Response · Authors · 2022-08-08
> **Response to Reviewer Pk6r**
>
> Dear Reviewer Pk6r:
>
> Thanks for your precious review time and constructive comments. We have added corresponding responses and results to cover your concerns. Please let us know if you still have unclear parts about our work. Looking forward to your quick feedback.
>
> Best,
>
> Paper4278 Authors

---

> > ### Comment · Reviewer_Pk6r · 2022-08-08
> > **Thanks for the response**
> >
> > Thanks for your efforts, I'm very glad to the see further discussion and extra comparisons which addressed my concerns. Thus, I will raise my rating to 6 weak accept afterwards. Hopefully, authors could include these in revision.

---

> > > ### Author Response · Authors · 2022-08-08
> > > **Re: Response to Reviewer Pk6r**
> > >
> > > Thanks for your helpful suggestions and positive response! We have added extra comparisons in our supplementary materials and will include all disscusions in our revision.

---

### Official Review · Reviewer_xCjf · 2022-07-12

**Rating:** 6
**Confidence:** 4
**Soundness:** 3 good
**Presentation:** 3 good
**Contribution:** 3 good

**Summary:**

To tackle the one/few-shot generative domain adaptation task, this paper proposes a novel method called DiFa, in which three different losses are introduced to achieve faithful adaptation. Among these three losses, the CLIP loss is computed between the embedding of the reference image and the embedding of the source domain for global-level adaptation. Attentive Style loss is calculated on the intermediate layer of the CLIP image encoder for local-level adaptation. Selective cross-domain consistency loss is introduced to select and retain domain-sharing attributes in the editing latent W+ space to inherit the diversity of the pre-trained generator. Extensive quantitative and qualitative comparisons show the superiority of the proposed method.

**Questions:**

See Weakness, can the authors provide domain adaptation results using aligned target images.

**Limitations:**

Adequate

**Strengths And Weaknesses:**

Strengths:
- The paper is well written and easy to follow. The framework itself is clear.
- The visualization results demonstrate the proposed method can generate high-quality results on diverse target images, which seems promising to me. The ablation study clearly shows the effect of each loss in the pipeline.

Weaknesses:
- The results for Row 3, 5, and 7 in Figure 5 look weird. Please clarify if an alignment step is used on the target image before domain adaptation. This is somewhat unfair to competing methods if the alignment step is not done.
- The faces generated by DiFa look almost the same except for the face shape. The proposed method appears to have failed to preserve facial identities.

---

> ### Author Response · Authors · 2022-08-02
> **Response to Reviewer xCjf**
>
> Thanks for your constructive comments.
>
> **The results for Row 3, 5, and 7 in Figure 5 look weird. Please clarify if an alignment step is used on the target image before domain adaptation.**
>
> For Fig. 5, most reference images are well aligned except for rows 4, 5, and 7 (row 1 represents the source images).
> We have conducted the comparison on the aligned images in rows 4, 5, and 7, and the results are provided in Fig. 18 of the revised supplementary materials.
> We observe that alignment has little impact on DiFa, StyleGAN-NADA, and Few-Shot Adaption.
> Our DiFa still outperforms other approaches with aligned reference images.
> In contrast, Mind The Gap is sensitive to the alignment of reference images and fails to deal with cases involving a large domain gap.
> After alignment, Mind The Gap achieves improved results for the small-domain-gap cases (*e.g.*, "FFHQ $\to$ Mona Lisa" in the last row of Fig. 18), but still produces images with visible artifacts for the large-domain-gap cases (*e.g.*, "FFHQ $\to$ Pharaoh" in the third-to-last row of Fig. 18).
>
>
> **The proposed method appears to have failed to preserve facial identities.**
>
> In the selective cross-domain consistency loss (Eqn. (6)), we have introduced the hyper-parameter $\alpha$ to control the proportion of preserved attributes.
> As shown in Fig. 10 of the supplementary materials, we illustrate that our DiFa can keep more facial identities in the human face domain as the proportion $\alpha$ of preserved attributes increases.
> To preserve more facial identities, we increase the $\alpha$ from 0.6 to 0.8, and show the comparison in Fig. 21 of the supplementary material.
> From the figure one can see that, with $\alpha$=0.8, our method retrains more facial attributes which can help to distinguish different faces, while adapting representative domain-specific styles from the reference images.

---

### Official Review · Reviewer_4nF5 · 2022-07-26

**Rating:** 6
**Confidence:** 4
**Ethics Flag:** Yes
**Soundness:** 3 good
**Presentation:** 3 good
**Contribution:** 3 good

**Summary:**

This paper presents an idea for a one-shot generative domain adaptation method.  To this end, this paper proposed to minimize both global and local loss. For global loss, this paper exploits the difference between the CLIP embedding of a reference image and the mean embedding of source. Whereas, for local-level constraining, the paper employs attentive style loss.
To validate the idea, the experiments are conducted on multiple pre-trained GAN such as StyleGAN2 trained on FFHQ, StyleGAN-ADA trained AFHQ-Cat. Both the qualitative and quantitative experiments are performed with the baselines.




**Questions:**

how diverse were the members of the user study group? On average, how much did they take to complete the assessment?


**Limitations:**

One limitation is that this model relies on CLIP embedding, which is trained on a very large corpus.

**Strengths And Weaknesses:**

The paper is generally well written. The paper motivates sufficiently the need of zero-shot generative model and argures the drawbacks of the existing methods.  Also, the related works are discussed adequately and the methods are presented by the equations and diagram. Both the equations and diagram are easy to follow. The experimental results are also convincing to justify their claims.

---

> ### Author Response · Authors · 2022-08-02
> **Response to Reviewer 4nF5**
>
> Thanks for your constructive comments.
>
> **How diverse were the members of the user study group?**
>
> (i) For the diversity of participants, we mainly consider the factors of gender, age, and knowledge background.
> The detailed statistics are as follows:
> Gender: Male: 53.3%, Female: 47.7%
> Age: < 20: 23.3%, 20$\sim$40: 56.7%, >40: 20%
> Knowledge background: researchers and developers on vision and graphics: 33.3%, artists and painters: 36.7%, others: 30%. The following table shows the details of the diversity of our participants.
>
> (ii) As for time cost, each participant spent about 90 minutes for completing the assessment on average.
>
> | Gender |        |
> |--------|--------|
> | Male   | 53.3% |
> |  Female   | 47.7% |
> | **Age** |        |
> | <20| 23.3% |
> |  20~40| 56.7%   |
> |  >40| 20 %  |
> | **Background** |        |
> | CV&CG | 33.3% |
> |  Arts | 36.7%   |
> |  Other | 30%   |
>
>
> **One limitation is that this model relies on CLIP embedding, which is trained on a very large corpus.**
>
> Yes, our DiFa relies on the CLIP embedding to guide the global-level and local-level adaption.
> Specifically, pre-trained CLIP is beneficial to our DiFa from three aspects.
> - Firstly, it can be easily extended to zero-shot domain adaption with appealing results.
> CLIP models are trained with 400 million image-text pairs and can encode both image and text into a shared embedding space. Thus, our DiFa can leverage a language description to guide the global-level adaption.
> - Secondly, the CLIP embedding is only required during training, but not during inference, thereby having no negative effect on the speed of inference.
> - Thirdly, better local-level adaption can be attained by CLIP other than VGG models. In particular, our experiments show that using intermediate tokens from CLIP models makes adapted generator acquire more domain-specific attributes (see Fig.12 in the supplementary materials).

---

### Review · Ethics_Reviewer_rzA8 · 2022-08-15

**Recommendation:**

In my opinion, it would be helpful to include the information on the distribution of reviewers in the full paper (likely the appendix). If it's available, it could also be useful to include information on race, as well as information on how the reviewers were recruited.

This wasn't raised by reviewer 4nF5, but I think it could also be helpful to provide more complete information on how the survey was run. The ethics checklist says that full text of the instructions provided to reviewers was given in Section 4.3, but that section seemed to only provide a general description of the study. Instead, it would be helpful to include (again, likely in the appendix) full screenshots of the study as shown to participants, as well as the text that was used to instruct them. This could be helpful to confirm, for example, that reviewers weren't nudged towards preferring one set of images over another.

**Ethics Review:**

Reviewer 4nF5 asked about the diversity of reviewers and how long they took to complete the task on average.

---

### Review · Ethics_Reviewer_AbRH · 2022-08-15

**Recommendation:** See above.

**Ethical Issues:**

Yes

**Ethics Review:**

Reviewer 4nF5 flagged this paper for ethics review. It's unclear from their review the precise reasons. Nevertheless, I agree that this paper deserves more attention.

The authors begin their discussion of potential negative societal implications in the conclusion of Section `5`:

>Meanwhile, our DiFa also makes AI more accessible to the public. Users could leverage our method to create the artworks with any desired styles, even without adequate computing and data resources. Albeit our work may bring potential concerns on the probability of producing fake images, we believe that they could be well addressed with the development of DeepFake detection and proper protocols.

On my reading, this shows that the authors have begun the necessary discussion, but the sections require more elaboration. Simply leaving the penultimate sentence at "may bring potential concerns on the probability of producing fake images" leaves the reader asking some basic questions like, "why" and "how so?" Given that, as the authors note, one clear consequence of this work is that "DiFa also makes AI more accessible to the public," does that increase the likelihood that this approach could be used to facilitate deceptive interactions? Could the approach help individuals either impersonate public figures to influence political processes, or use the approach as a tool of hate speech or abuse?

The authors note that this might be mitigated "with the development of DeepFake detection and proper protocols," but offer no further elaboration.

Ultimately, this can be rectified. The paper simply requires more elaboration on both the potential negative societal impacts and the potential mitigations available. Were the authors to add a few sentences more on each topic, I believe the issues would be addressed.

---

### Meta-Review · Area_Chair_t7nF · 2022-09-05

**Recommendation:** Accept
**Confidence:** Less certain

**Metareview:**

The paper focuses on one-shot generative domain adaption and proposes a novel method to obtain faithful adaptation. It is well written and easy to follow. The visualization results and quantitative evaluations demonstrate the proposed method can generate high-quality results. In all, the meta-reviewer considers the contribution of this paper significant, and worth publication. The authors need to incorporate ethics reviews when preparing the camera ready version.

**Award:**

No

---

### Decision · Program_Chairs · 2022-09-14

Accept